# Microscopic insight into non-radiative decay in perovskite semiconductors from temperature-dependent luminescence blinking

Marina Gerhard[1], Boris Louis [1,2], Rafael Camacho [2], Aboma Merdasa[3], Jun Li[1], Alexander Kiligaridis[1], Alexander Dobrovolsky[1], Johan Hofkens[2] & Ivan G. Scheblykin [1]

Organo-metal halide perovskites are promising solution-processed semiconductors, however, they possess diverse and largely not understood non-radiative mechanisms. Here, we resolve contributions of individual non-radiative recombination centers (quenchers) in nanocrystals of methylammonium lead iodide by studying their photoluminescence blinking caused by random switching of quenchers between active and passive states. We propose a model to describe the observed reduction of blinking upon cooling and determine energetic barriers of 0.2 to 0.8 eV for enabling the switching process, which points to ion migration as the underlying mechanism. Moreover, due to the strong influence of individual quenchers, the crystals show very individually-shaped photoluminescence enhancement upon cooling, suggesting that the high variety of activation energies of the PL enhancement reported in literature is not related to intrinsic properties but rather to the defect chemistry. Stabilizing the fluctuating quenchers in their passive states thus appears to be a promising strategy for improving the material quality.

---

[1] Division of Chemical Physics and NanoLund, Lund University, Box 124, 22100 Lund, Sweden. [2] KU Leuven, Molecular Imaging and Photonics, Celestijnenlaan 200F, Box 2404, 3001 Leuven, Belgium. [3] Hybrid Materials Formation and Scaling, Helmholtz-Zentrum Berlin für Materialien und Energie GmbH, Helmholtz-Zentrum Berlin, Albert Einstein Strasse 16, D-12489 Berlin, Germany. Correspondence and requests for materials should be addressed to M.G. (email: marina.gerhard@chemphys.lu.se) or to I.G.S. (email: ivan.scheblykin@chemphys.lu.se)

rgano-metal halide perovskites (OMHPs) have attracted a lot of interest as promising materials for light emitting and photovoltaic devices[1]. Although processed from solution, OMHPs possess good carrier transport and often high luminescence yield attributed to their defect tolerance[2–4]. However, under typical solar flux illumination, the photoluminescence (PL) quantum efficiencies of perovskite materials are still far from unity[5,6]. As a consequence, different methods have proved useful to increase the PL yield by reducing non-radiative decay, for instance light soaking[7,8], surface passivation[9–11], and controlled exposure to moisture[12]. Further evidence for non-radiative decay is provided by the temperature-dependent PL quantum yield, which increases upon cooling[13–20], as in most luminescent solids[21,22]. Dependence of non-radiative recombination on local properties becomes apparent in the spatially inhomogeneous PL[7,9,23,24]. Some of these PL quenching channels are of extraordinary efficiency (therefore called 'super traps') leading to PL blinking observed in individual crystals and films[23,25–30]. Obviously, there are several sources of non-radiative recombination which are, to a large extent, unexplained. Thermodynamics tells us that to go above the current record level of 22.7% of solar cell efficiency[31] towards the Shockley–Queisser limit, non-radiative channels must be eliminated from the material. Therefore, understanding their fundamental origin is absolutely crucial[32].

To obtain insight into non-radiative recombination processes, we can imagine a thought experiment where we are able to monitor each act of charge trapping and recombination. Each quenching site can either be empty or filled by a charge or passivated in any other way (e.g. chemically). Therefore, the quencher fluctuates over time between an active state (able to accept a charge) and a passive state (not available for charges). Time-averaging of this random process gives us the effective rate of non-radiative recombination via the particular trap.

To see beyond the ensemble-averaged picture, one would like to study a small volume of a semiconductor, where only one non-radiative recombination channel (e.g. a deep trap) is present. The tool for that is single molecule spectroscopy, which studies individual entities one by one, whereas the ensemble of these entities represents the bulk. PL from a small volume of a semiconductor often possesses large fluctuations (PL blinking, intermittency or flickering) at time scales from microseconds to minutes and above[23,25,27,33,34]. This means that the non-radiative decay in the particle undergoes large fluctuations due to, for example, a few non-radiative centers switching between their active and passive states. PL blinking has also been observed in quite large objects such as perovskite microrods[23], microcrystals[30], and graphene oxide layers[35], where concentration and spatial distribution of the non-radiative centers can be assessed by luminescence microscopy either directly or using optical super-resolution methods[23,25,36]. As fluorescence blinking of individual molecules allowed for breaking of the diffraction limit in optical microscopy[37], fluctuating quenching processes in a semiconductor can potentially allow for resolving the properties of individual recombination channels that otherwise are convoluted in the ensemble-averaged rates. As we will show, OMHP semiconductors are ideal systems to realize such a desired experiment in practice.

The goal here is to communicate a mechanistic view on the non-radiative processes in semiconductors using $CH_3NH_3PbI_3$ (MAPbI$_3$, MA = $CH_3NH_3^+$) perovskites as a model system. Studying of PL blinking enables us to estimate the quencher concentration, their efficiency and temperature-dependent properties. In particular, we can track the properties of individual quenchers as function of temperature, which has, to the best of our knowledge, not been done before. We show that switching of the non-radiative channels from the active to passive state is temperature activated with thermal barriers in the range of 0.2–0.8 eV, which is consistent with the range of estimated activation energies for ion migration[38–41]. We demonstrate that non-radiative recombination reduces due to the decreasing efficiency of the active quenching sites and potentially also due to decreasing of their concentration at low temperature. Partial hindering of charge diffusion at low temperatures as well as the presence of activation barriers in trapping and recombination processes will be discussed. The quenching sites must be related to a particular local environment (immobile chemical defect or physical defect) and possess highly individual temperature dependencies of their activity, which cannot be directly explained by an exponential activation law.

## Results

**Time-integrated PL intensity**. The PL images recorded at different temperatures (Fig. 1a–d) reveal remarkable differences, demonstrating that the temperature dependencies of PL intensities of different crystals differ drastically. While at room temperature the crystal appearances vary from bright to invisible, cooling down increases the overall PL intensity (Fig. 1e) and makes the PL intensities of individual crystals more similar to each other (Fig. 1f). The observed enhancement of the ensemble averaged PL intensity by two orders of magnitude upon cooling down to 77 K is consistent with the literature[13–19]. Using quantitative measurements of luminescence brightness[42] for several individual crystals of known size and known local excitation power density, we estimate their PL quantum yield at room temperature to be around 1% and more than 20% at 77 K (see Supplementary Notes 1–3). The $PL(T)$ data were fit with the Arrhenius model[22]

$$PL(T) = I_0 \frac{k_{\mathrm{r}}}{k_{\mathrm{r}} + k_{\mathrm{nr}}} = \frac{I_0}{1 + \frac{k_{\mathrm{nr}}}{k_{\mathrm{r}}}} = \frac{I_0}{1 + A \cdot \exp\left(-\frac{\varepsilon}{k_{\mathrm{B}}T}\right)} \quad (1)$$

Where $k_{\mathrm{nr}}$ and $k_{\mathrm{r}}$ are the non-radiative and radiative recombination rates, $I_0$ and $A$ are free parameters and $\varepsilon$ is the effective activation energy and $T$ is temperature. The fit yielded an activation energy $\varepsilon$ of 63 meV, which is in the upper range of previously reported values of 68 meV[19], 62 meV[14], 32 meV[15] and 19 meV[20].

The PL intensity of individual crystals (Fig. 1g–j) shows very distinct temperature dependence, which is generally reproducible for cooling and subsequent heating. In most cases, the slope cannot be described by a simple activation law. To illustrate this, we fitted the temperature-dependent intensity of a group of 25 crystals with Eq. 1 (see Fig. 1g–j and Supplementary Figures 1–2). The criteria of the selection procedure are described in Supplementary Method 1. Figure 1k shows the resulting distribution of the parameter $\varepsilon$, which is centered around a value of about 0.13 eV and spreads over an energy range of more than 0.1 eV. The peak energy of this distribution deviates from the fitting result in Fig. 1e. We attribute this to the fact that only a subgroup of the overall ensemble was selected to perform individual fits, whereas the ensemble averaged PL intensity was obtained from integrating the intensity counts of the whole image, including also the crystals which are invisible at room temperature but become visible at low temperature.

Our observations suggest that the PL intensity of individual nanocrystals is controlled by a few quenchers with different activation barriers. Peculiarities in the temperature dependent PL quantum yield of nano- and micrometer sized objects have been reported before and were attributed to local variations of the phase transition temperature due to the presence of defects, leading to a local maximum of the PL quantum yield at temperatures around 140 K[24]. However, although such effects

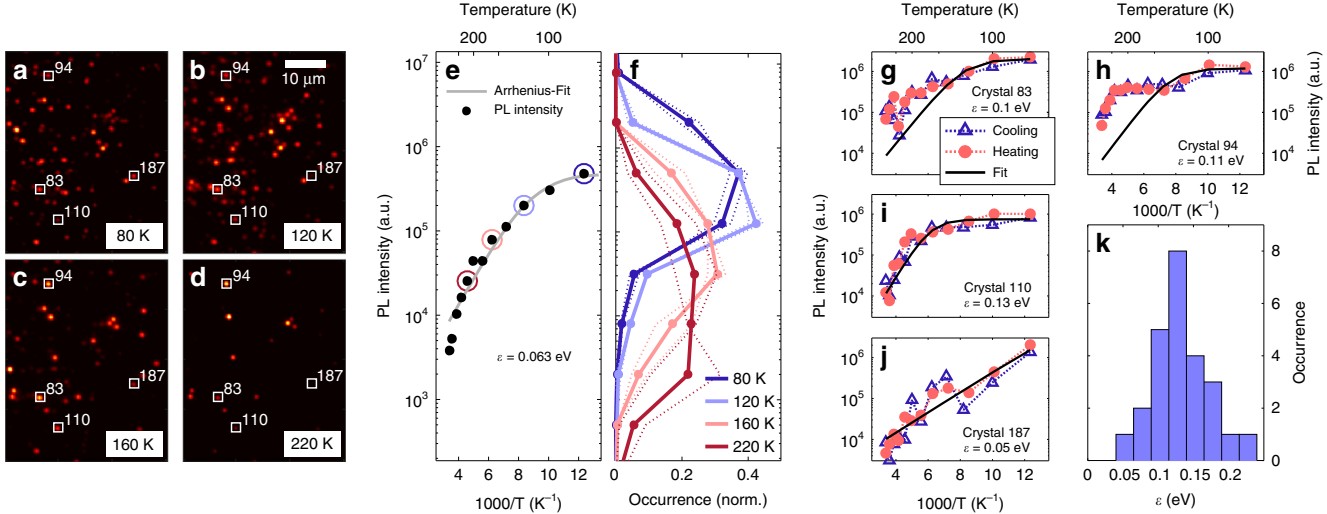

**Fig. 1** Temperature dependent PL intensity. The PL images in **a**–**d** are the time-averages of the movies recorded over 100 s with 50 ms exposure time of the same sample region at temperatures of 80, 120, 160 and 220 K, respectively. Several crystals in this region are marked. Panel **e** shows the overall PL intensity of this region, for which the signal was integrated over all pixels of the image and averaged over 100 s of the movie recording time. The gray line represents an Arrhenius-fit giving an activation energy $\varepsilon$ of 0.063 eV. The circled data points indicate some selected temperatures, for which we show the corresponding distributions of the PL intensities of individual crystals in **f**. Panels **g**–**j** show the temperature-dependent PL intensity of the four individual nanocrystals, which are marked in **a**–**d**. We fit this data with Eq. (1) to demonstrate strong deviations from the Arrhenius activation law. More fitting examples can be found in Supplementary Figure 2. Panel **k** shows a distribution of the apparent activation energies $\varepsilon$ obtained by fitting **PL(T)** of these 25 crystals. The sample was excited at 458 nm (CW) with an average excitation power density of 0.3 W cm$^{-2}$ (for calculation of the excitation power density, see Supplementary Note 2). The data presented in Figs. 2–4 was recorded under the same excitation conditions

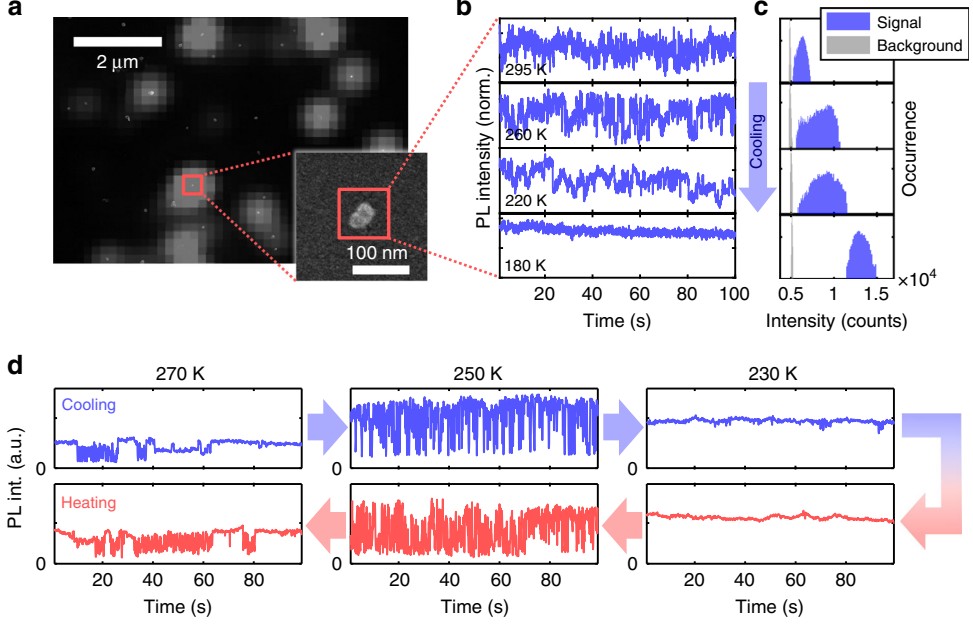

**Fig. 2** Exemplary blinking data. Overlapped SEM and PL microscopy images and a detailed image of one of these crystals are presented in **a**, for more SEM data and more detailed analysis see Supplementary Note 1. From quantitative analysis of this data we estimate an average crystal diameter of 70 nm. Panel **b** shows normalized temperature-dependent PL intensity traces extracted from this crystal. Panel **c** summarizes the corresponding intensity histograms with logarithmic vertical axis. Panel **d** shows intensity traces recorded for heating and cooling of another crystal, exemplifying that the blinking characteristics are reproducible and very sensitive to a small change of temperature. All plots in **d** are shown with the same intensity scale

seem to be present in the crystals investigated here (see e.g. Figure 1j), they cannot be the only source of irregularities in the temperature dependent PL quantum yield. Direct evidence for the influence of individual quenching sites is also given by the pronounced blinking behavior[23,25–30]. Below we will characterize the temperature dependence of blinking in order to establish its connection to the PL quantum yield.

**PL intensity fluctuations.** As illustrated in Fig. 2a–c, the blinking process becomes less pronounced upon cooling. A further typical observation was a reproducible drastic change of the blinking behavior within a particularly narrow temperature range of only about 20 K for cooling and subsequent heating (see Fig. 2d).

Due to the combined effect of several fluctuating quenchers per crystal, the blinking traces observed here show very complicated

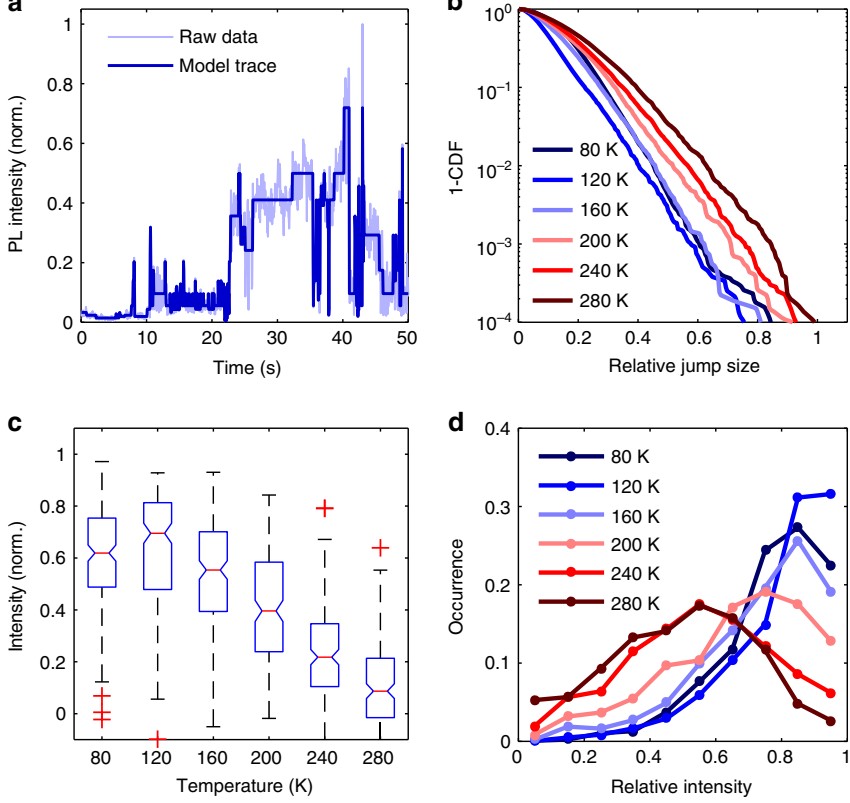

**Fig. 3** Analysis of the relative PL intensity fluctuations. To make the blinking amplitudes comparable, we divided each model trace by its maximum and extracted relative blinking amplitudes ranging between 0 and 1. Panel **a** shows an intensity transient recorded at 240 K and the corresponding model trace, from which jump events were extracted. One minus the cumulative distribution function (1—CDF) of the relative blinking amplitudes is presented in **b**. The function 1—CDF($x_0$) gives the probability for x to be larger than $x_0$. For example, at 280 K the probability for the relative jump size to be more than 0.4 is 0.2. The box plot in **c** shows the distributions of the lowest relative intensity values. The central marks indicate the median of the distribution and the upper and lower limits of the boxes correspond to the 25th and the 75th percentiles of the respective intensity distributions. The length of the whiskers corresponds to the intensity value, which is closest to the 1.5-fold box size. Data points exceeding this range are considered as outliers (red crosses). Panel **d** shows the occupation probability of intensity levels calculated from the intensity normalized model traces

behavior. Thus, a conventional analysis of "on" and "off" times, as often performed for quantum dots[43,44], does not appear to be a good approach in our case. For an unbiased analysis of the intensity jumps (*blinking amplitudes*) and the blinking dynamics, we applied a Bayesian method to detect stable intensity levels[45] and an additional procedure to detect short blinking events (see Methods). By this we replaced each experimental PL transient by a noiseless model transient (Fig. 3(a)) which was further analyzed.

In Fig. 3b, we show the distributions of relative intensity jumps extracted from the intensity-normalized blinking transients for a given temperature by plotting their cumulative distribution function (CDF). Although there is a ten-fold increase of the absolute blinking amplitudes upon cooling (see Supplementary Figure 3), the overall fraction of PL that can be quenched by individual non-radiative channels decreases. The reason for this is that the enhancement of the blinking amplitudes is less than the enhancement of the maximum PL intensity, which is also indicated by the distribution of the lowest intensity levels at a given temperature in Fig. 3c. However, the long whiskers and outliers in the box plots indicate that also large intensity jumps (nearly complete quenching) are still observed in some crystals at low temperature. The overall probability distribution to find the PL intensity at a certain value relative to the maximum intensity is presented in Fig. 3d. At high temperature the crystal intensity resides preferentially at intermediate intensity levels, from where it can blink up or down. This is contrary to low temperature,

where the crystals spend most of their time at bright intensity levels.

We also analyzed the time periods, during which the PL resides at a particular intensity level (see Supplementary Figure 4). As suspected from Fig. 3d, we found that at low temperatures, the residence times at bright intensity levels are longer than those at dim intensity levels. This difference, however, vanishes at high temperature.

The analysis of the blinking amplitudes showed that upon cooling, there must be a growing contribution to the PL signal which, for some reason, cannot be quenched by the fluctuating quenchers. We checked if saturation (trap filling) of the quenchers could reduce the blinking amplitudes at low temperature, however, we found no evidence for decreasing blinking amplitudes after increasing the excitation power (see Supplementary Note 4). We will elaborate on the origin of this non-quenchable PL in the discussion section.

To characterize the switching processes, we measured the switching of the quenchers by counting the blink events. For many crystals (see Fig. 4 and Supplementary Figure 5), we observe a clear reduction of the switching rate with decreasing temperature, however, in other cases (see e.g. crystal #95) blinking is also present at low temperatures and the switching rate shows a non-monotonous temperature dependence. In the data we find no obvious correlation of the switching rate with the phase transition temperature of MAPbI$_3$ and the observation of

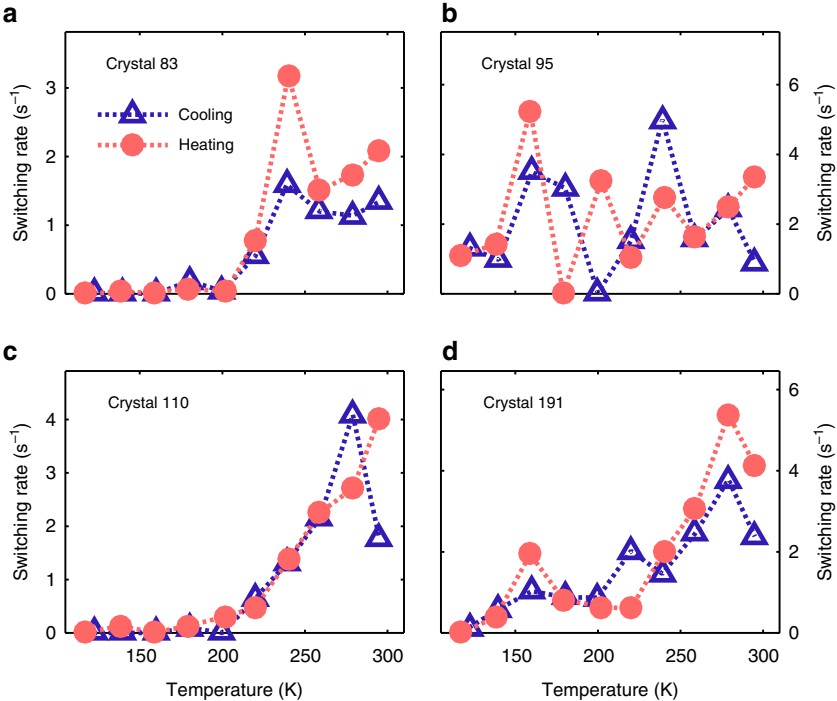

**Fig. 4** Analysis of blinking dynamics extracted from the model traces. For calculation of the switching rates, all blink events with amplitudes larger than 20% of the mean intensity of a PL transient were considered. The number of events was counted and dividing by the measurement time of 100 s yielded the switching rates presented in this figure. The switching rates of the individual crystals presented here in panels (**a-d**) demonstrate distinct and reproducible temperature dependencies, for more examples see Supplementary Figure 5

blinking in the low temperature regime suggests that in principle both the tetragonal and the orthorhombic phase can incorporate quenching sites that induce PL blinking. Remarkably, the observed temperature dependencies are reproducible, as they are detected in both the heating and the cooling cycle. Such reproducibility has already been noted in relation to the PL intensity in Fig. 2b.

**Mechanistic view on the observed temperature dependencies**
In order to account for the fluctuations of the experimentally observed PL quantum yield $\Phi(T,t)$ and its temperature dependence, we use the classical assumption that $\Phi$ is determined by the competition between radiative and non-radiative decay rates of the excited state. In order words we assume that the quenching is dynamic. The non-radiative recombination rate of an individual crystal is the sum of all non-radiative rates. We split them into two parts: A constant rate $k_c$ and a time-dependent randomly switching rate $k_q(t, T) = k_{q,0} \cdot n(t, T)$, where $k_{q,0}$ is the decay rate per quencher and $n$ is the number of active quenchers out of a total number of $N$ quenchers at time $t$. Each quencher $i$ can be either in an active ($Q_i = 1$) or passive state ($Q_i = 0$). With this, we can give the following expression for $\Phi(T,t)$:

$$\Phi(T, t) = \frac{k_r(T)}{k_r(T) + k_c(T) + k_q(t, T)} = \frac{\Phi_0(T)}{1 + \kappa(T) \sum_{i=1}^{N} Q_i(T, t)}$$
$$= \Phi_0(T) \Phi_{\text{switching}}(T) \tag{2}$$

where $\kappa(T) = k_{q,0}(T) / (k_r(T) + k_c(T))$ is the ratio of the decay rate via one quencher being in active state to the sum of all other rates and $\Phi_0(T) = k_r(T) / (k_r(T) + k_c(T))$ is the PL quantum yield in the absence of the switching non-radiative channel. While the time-dependent PL intensity fluctuations are determined by $\kappa(T)$ and $Q_i(t)$, the overall time-averaged PL intensity is also

determined by the pre-factor $\Phi_0(T)$. The idea of a fluctuating non-radiative rate in Eq. 2 has already been used before to rationalize PL blinking of semiconductor quantum dots (model of Multiple Recombination Centers (MRC))[46]. Note that here we assume that the quenching rate $k_{q,0}$ is equal for all fluctuating quenchers, which allows us to define $\kappa(T)$ as a simple modeling parameter. It turns out that this simple approach is sufficient to describe the experimentally observed trends, however, in real crystals it is more likely that the quenching rates of the fluctuating quenchers actually reveal a certain distribution, which is not known so far.

We can roughly estimate $\kappa(T)$ from the experimental data using the temperature dependence of the maximum relative blinking amplitudes (for details see Supplementary Method 2). $\kappa(T)$ is approximately 10 at room temperature, meaning that the non-radiative rate induced by only one fluctuating quencher being in the active state is about 10 times larger than the sum of all other non-fluctuating decay rates in the system. Due to this high ratio, we observe very efficient quenching of the fluctuating quenchers at room temperature with large relative blinking amplitudes. Upon cooling, $\kappa(T)$ must decrease to explain the experimentally observed decrease of the relative PL blinking amplitudes.

To calculate the contribution of the switching processes to the overall PL quantum yield, we need to average the fluctuating part of Eq. 2 over time. Averaging over time of $\sum_{i=1}^{N} Q_i(T, t)$ gives an averaged number of active quenchers at a given temperature which, if temperature dependent, is important for the overall PL enhancement upon cooling together with $\kappa(T)$.

We propose that transitions of the quencher from the passive to the active state and vice versa occur via an energetic barrier (see Fig. 5 a–b). The transfer rate from passive to active state ($k_{p \to a}$) and from active to passive state ($k_{a \to p}$), which are equal to

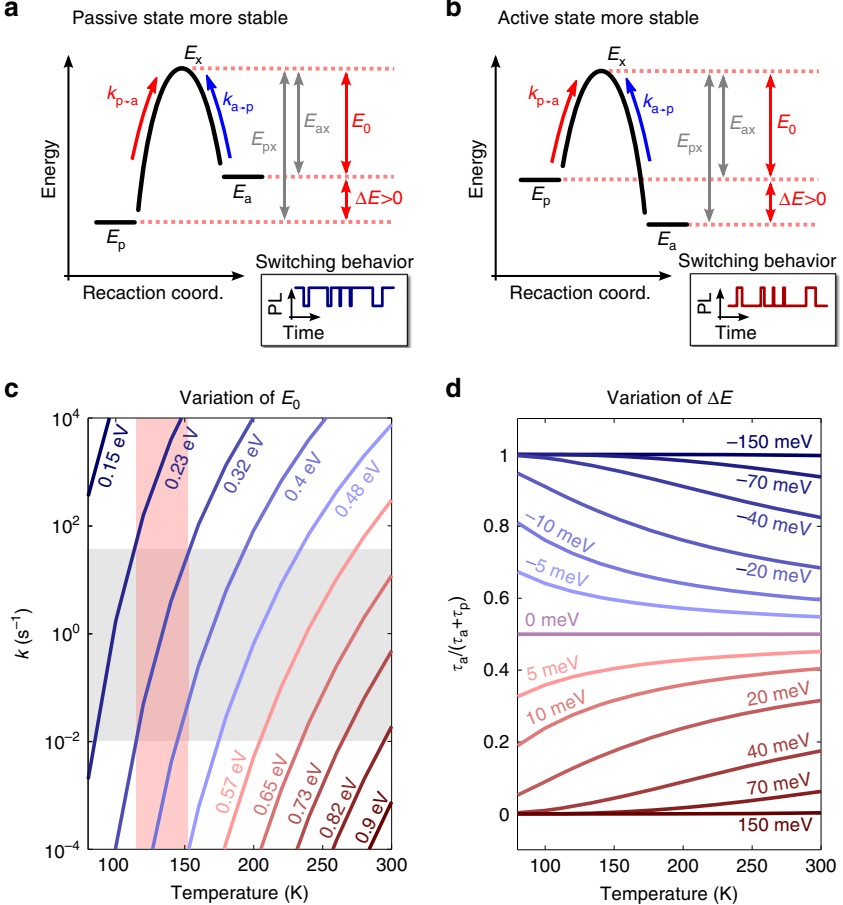

**Fig. 5** Proposed model for switching of the quencher between the active and passive state. Panel **a** sketches the possible energetic arrangements between active and passive states. The states are characterized by their energies $E_a$ and $E_p$, respectively and the energy level of the barrier $E_x$. The passive state can be either energetically below (left panel) or above (right panel) the active state. In order to switch between the two metastable configurations (active and passive), the quencher needs to overcome an energetic barrier $E_{ax} = E_x - E_a$ or $E_{px} = E_x - E_p$ depending on in which state (active or passive) the system stays initially. Depending on which of the two barriers is higher, the switching pattern is different, i.e. long passive times in case of an energetically favored passive state (left panel) and long active times when the active state is lower in energy (right panel), as shown in the schematic blinking transients in the insets. **b** Switching rate **k** as function of temperature for different activation barriers $E_0 = min(E_{ax}, E_{px})$. Here, we calculated **k** according to Equation 3 (note that $E_0 = E_{ax} = E_{px}$ for $\Delta E = 0$). The grey area indicates the range of switching rates that can be detected experimentally. The upper limit is given by the measurement time and the lower limit by the integration time. For example a quencher with $E_0 = 0.32$ eV gives blinking observed from 120 to 150 K, as highlighted in red. **c** Fraction of time the quencher spends in the active state for different temperatures and energetic spacings $\Delta E$ between active and passive state calculated from Equation 4

the inverse times needed to overcome this barrier (passive and active residence times, $\tau_{active}$ and $\tau_{passive}$) can be calculated as follows:

$$k_{p \to a} = \frac{1}{\tau_{passive}} = k_0 \cdot \exp\left(-\frac{E_{px}}{k_B T}\right)$$
$$k_{a \to p} = \frac{1}{\tau_{active}} = k_0 \cdot \exp\left(-\frac{E_{ax}}{k_B T}\right)$$
(3)

Here, $E_{ax} = E_x - E_a$ and $E_{px} = E_x - E_p$ are energy barriers for switching to the active ($E_a$) and passive ($E_p$) states, respectively, via the energetic barrier $E_x$ (see Fig. 5 a–b for details). The rate $k_0$ is an attempt frequency, here assumed to be on the order of $10^{12}$ s$^{-1}$, corresponding to the typical time scale of lattice vibrations[39]. For further discussion, it is convenient to define the parameter $E_0 = \min(E_{ax}, E_{px})$, which is the minimum barrier height for switching between the active and passive configuration and the energetic difference between the states $\Delta E = E_a - E_p$. Both parameters are presented in the scheme in Fig. 5a–b.

Let us first estimate the range of the barrier height $E_0$ for which we would expect to observe PL intensity fluctuations at our experimental conditions. To test the influence of $E_0$ on the switching times, we use Equation 3 to calculate the switching rate $k$ as function of temperature for different $E_0$, assuming $\Delta E = 0$ for simplicity. From the plot in Fig. 5c, it becomes obvious that the temperature dependence of the switching rates is very strong. The grey area in the plot shows the range of experimentally detectable rates. The temperature interval over which the quencher with particular $E_0$ resides in the detection range is about 50 K. Beyond this range, the switching becomes either too fast or too slow to be observed. For example, a quencher with $E_0 = 0.32$ eV can give PL blinking at temperatures from 120 to 150 K only. Therefore, several quenchers with different barrier heights are necessary to explain the observation of blinking over the broad temperature range between 80 and 295 K. Figure 5(c) shows that the range of barrier heights $E_0$ for which we expect to observe switching in the studied temperature range is limited to ~0.2–0.8 eV.

Note that the switching rates we are discussing here are very slow in comparison to the attempt rates. The probability of

jumping over a barrier of 0.8 eV is indeed very small at room temperature ($k_BT = 0.025$ eV). However, due to the high attempt rate $k_0$, the system has $10^{12}$ attempts per second to overcome this barrier. Employing Equation 3 for $E_0 = 0.8$ eV yields a switching rate of 0.01 s$^{-1}$. Thus, despite the high barrier it is still likely to observe a blink event during the measurement time of 100 s. In the framework of our model, the range of barrier heights is directly connected to the crystal re-arrangements at the atomic scale.

It is important to make a distinction between the activation barrier for switching $E_0$ of the quencher (microscopic parameter) and the experimental phenomenological activation parameter $\varepsilon = 0.063$ eV obtained from the Arrhenius fit of the temperature dependent PL intensity (Fig. 1). In order to explain the parameter $\varepsilon$, one needs a detailed physical model of the microscopic processes. In the model proposed here, $\varepsilon$ does not correspond to any energy difference between real states. It appears from a convolution of several temperature dependent factors: Switching of the fluctuating quenchers (barrier energies ranging between 0.2 and 0.8 eV), efficiency of the quencher described by $\kappa(T)$ and persistent quenching characterized by $\Phi_0(T)$.

The difference $\Delta E = E_p - E_a$ gives the energy shift between the active and passive states of the quencher. If $\Delta E$ is different from zero, the quencher will spend more time in the energetically lower state. Let us calculate the fraction of time the system spends in the active state as function of $\Delta E$. Using Eq. 3, we can write:

$$\frac{\tau_{\text{active}}}{\tau_{\text{active}} + \tau_{\text{passive}}} = \frac{1}{1 + \frac{\tau_{\text{passive}}}{\tau_{\text{active}}}} = \frac{1}{1 + \exp\left(\frac{\Delta E}{k_B T}\right)} \quad (4)$$

The fraction of time spent in the active state is plotted in Fig. 5d for different $\Delta E$. The plot demonstrates that the temperature dependent ratio of the active versus passive times is rather weak in our model. For energy differences much larger than $k_B T$, corresponding to $|\Delta E|$ above 0.1 eV, the quencher spends most of the time either in the active state ($\Delta E$ negative) or in the passive state ($\Delta E$ positive). Therefore, to cause fast blinking, a quencher must have a rather small $|\Delta E|$.

In order to test the validity of our model, we employed Monte-Carlo simulations based on Eqs. 2 and 3 (see Methods section and

Supplementary Method 2 for details). Figure 6(a) shows an example of the simulation together with the experimental data for one of the crystals. Indeed, the experimentally observed features of the irregular PL enhancement (Fig. 6c), switching rate (Fig. 6d), and general appearance of the PL transients can be semi-qualitatively reproduced with several quenchers (five in the case shown). The simulated PL enhancement presented in Fig. 6c comes exclusively from the fluctuating quenchers, because so far the pre-factor $\Phi_0(T)$ was considered to be 1 and temperature independent. The experimentally observed quenching at elevated temperatures is stronger than obtained in the simulations, which can be rationalized by the presence of the constant non-radiative channel $k_c$ leading to temperature dependent $\Phi_0(T) < 1$, which could e.g. be induced by the crystal surface.

## Discussion

The most popular systems possessing PL blinking are small (3–10 nm) semiconductor nanocrystals or quantum dots (QD). The widely accepted physical mechanism behind PL blinking in these single quantum systems is switching between a neutral (emissive) and a charged state (non-emissive, due to Auger recombination) via charge tunneling. Tunneling explains the absence of a clear temperature dependence of QD blinking[33,34,43,44]. However, in the case of MAPbI$_3$ nanocrystals, the switching reveals very distinct temperature dependencies, suggesting that temperature plays an important role in the process. Moreover OMHP perovskite crystals possessing PL blinking are many orders of magnitude larger than QDs (50–1000 nm)[23,25,36]. PL blinking in MAPbI$_3$ is observed at relatively low excitation power density (see Supplementary Note 2), suggesting that Auger processes, which are likely to cause blinking in QDs, can be neglected[23]. We propose that PL blinking in MAPbI$_3$ and other OMHPs occurs due to the structural dynamics in these rather soft materials. This dynamics lead to fluctuations of the non-radiative recombination rate (switching of individual quenchers between active and passive state) at time scales from milliseconds to minutes and even longer. Presence of just a few randomly fluctuating quenchers per crystal leads to notable PL fluctuations, which are not masked by averaging

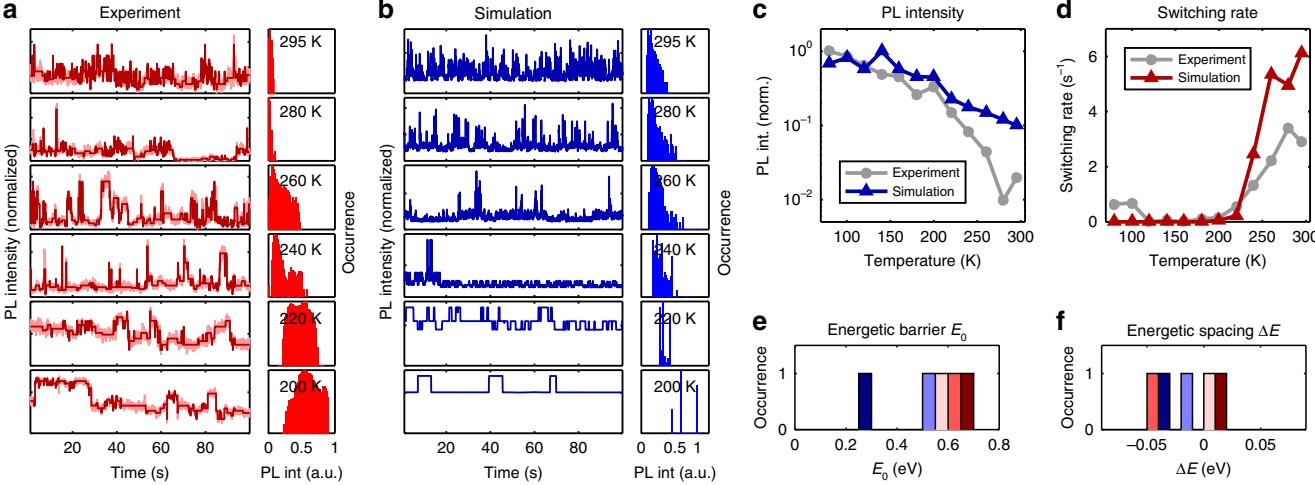

**Fig. 6** Simulated blinking transients and comparison to the experimental data for one crystal (#110). Panel **a** shows normalized experimental intensity transients recorded at various temperatures with the corresponding intensity histograms in the right column. The dark red curves were created from the raw data using the Bayesian method (see Methods) and served for further analysis of the blinking behavior. In panel **b** we show simulated intensity transients created from our model. Here we chose five quenchers and adjusted the energies of the individual quenchers **E$_0$** and **ΔE**, such that the resulting intensity curves mimic the experimental observations for this crystal (see Supplementary Figures 6–8 for more examples). Panels **c** and **d** show the PL intensity and the switching rate (number of switches per second) extracted from the experimental and simulated data. In order to provide an overview over the energetics of the simulated quenchers, we plot histograms of the minimum energetic barrier **E$_0$** and the energetic spacing $\Delta E = E_p - E_a$ in panels **e** and **f**, respectively

effects. The mathematics behind this is described in the section above and it is conceptually the same as in the Multiple Recombination Centers (MRC) model earlier proposed for QDs[46].

Temporal fluctuation of the PL allowed us to see contributions of individual quenchers to the PL quantum yield of MAPbI$_3$ nanocrystals. According to the simulations, in order to reproduce the experimental observations, we need on average about one to three active quenchers per crystal (see Supplementary Figure 9). By assuming a volume of $70 \times 70 \times 70$ nm$^3$, this allows us to estimate an average quencher (or trap) density of $1.6 \times 10^{16}$ cm$^{-3}$. This defect density is well comparable to other reports for similarly prepared polycrystalline films[18,47]. Note though that PL blinking can reveal fluctuating non-radiative channels only, whereas persistent quenching e.g. surface recombination cannot be detected in this way. The PL decay of the studied crystals showing a long tail of about 50 ns at room temperature (see Supplementary Note 5) serves as a very rough indication of the concentration of non-radiative centers and shows that our samples are not different from usual polycrystalline films with small grain sizes[18,48] where the concentration of traps influencing the PL is usually estimated to be in the range of $10^{15} - 10^{16}$ cm$^{-3}$. Matching of these numbers strongly supports the idea that we indeed resolved most of the quenchers due to their fluctuations.

So far, the origin of the intensity fluctuations remains an open question. Today, a vast amount of literature on OMHPs reports on numerous phenomena like hysteresis, ferroelectricity, light soaking, phase segregation, self-healing, etc. related to local unit cell re-arrangement, chemical composition change and diffusion of ions, their vacancies and other species[49–51]. Literature contains a broad variety of estimated or calculated activation energies for ionic motion in MAPbI$_3$ (for iodine 0.05 to 0.58 eV, hydrogen 0.17–0.5 eV, lead 0.80–2.31 eV and methylammonium 0.80–0.84 eV)[38,39,41,52]. From these numbers we can conclude that the iodine anion is the first candidate to be responsible for structural changes in MAPbI$_3$ due to relatively low activation barriers[53,54]. Also protons emerging from the residual water or deprotonation of MA should be considered as highly mobile defects[40,41]. The

overall range of the reported energies of ion migration matches nicely the range of energy barriers from 0.2 to 0.8 eV that we estimated for switching of the quenchers. Thus, we suggest that ion migration is most probably involved in the formation and annihilation (switching) of the quenchers in MAPbI$_3$.

Note that some studies showed evidence that ion migration can be photo-activated[40,53]. By varying the excitation power, we tested if photoactivation becomes also apparent in the switching behavior of the nanocrystals studied here but did not observe any clear effect of excitation power on the blinking characteristics (see Supplementary Note 4). The reason for this could be that the excitation densities applied here are actually higher than the range of densities in which photoactivated ion conductivity has been observed experimentally[26,40]. Moreover it is not known if the ions contributing to photo-activated ion conductivity and the ones inducing switching of the quenchers are the same.

We stress that the individual quenchers causing PL blinking must be quite extraordinary in terms of the introduced non-radiative rate. Indeed, to result in large fluctuations of the PL, the non-radiative rate introduced by a single quencher must be comparable with the sum of all recombination rates existing in the crystal before the quencher was activated. A quencher consisting of a single trap state must have the energy level close to the middle of the band gap to provide the highest non-radiative recombination rate. Recently it was pointed out that certain oxidation states of interstitial iodine[52,55] and its vacancy[56] can act as such efficient quenchers. However, most of the abundant defects in OMHPs reveal energy levels close to the band edges and therefore they are not able to quench PL efficiently. On the other hand, as suggested by some of us[23], efficient quenching could also arise from a complex of several defects, which are otherwise harmless individually. Such a complex can provide an efficient non-radiative decay channel by creating several energy levels inside the band gap[57,58]. For example a complex between a shallow electron and shallow hole trap can work as efficient non-radiative center (Fig. 7a).

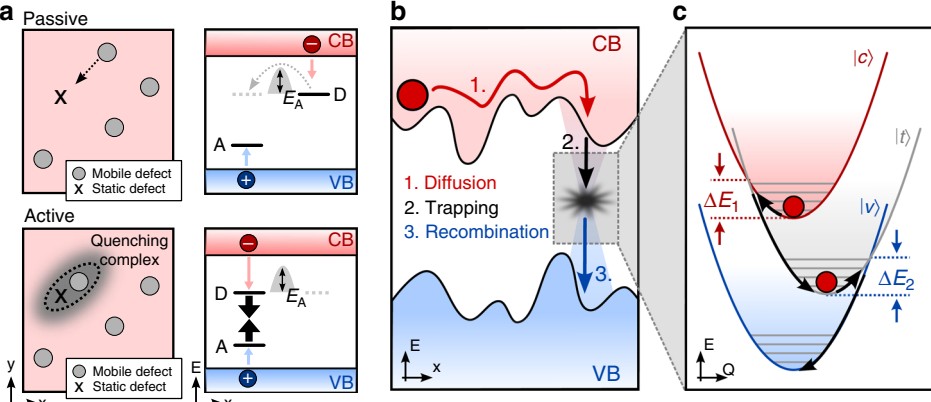

**Fig. 7** Proposed switching and quenching mechanisms. The mechanism of switching in **a** is exemplified for the formation of a donor-acceptor complex in real space (left) and an energy level diagram (right). CB denotes the conduction band, VB the valence band and D (A) the donor (acceptor) level inside the band gap. The system contains at least two types of defects and one of these defects is mobile. Both species on their own do not act as efficient quenchers (see upper panels: Passive configuration), but when two defects of different kind get close, they form an active quencher with a high quenching efficiency (lower panels). Panel **b** is a schematic of the non-radiative decay process in a semiconductor. It consists of diffusion (1), trapping (2) and recombination (3). Each of these steps could be thermally activated. Diffusion to the quenching site can be limited due to a disordered lattice potential. Energetic barriers for trapping and recombination arise due to reorganization of the nuclei of the local center, as illustrated in **c**. The system can be in three different electronic states and the corresponding potentials as functions of the nuclear coordinates of the center are shown: Excited state (state $|c\rangle$ = free electron in the conduction band, neutral center), intermediate state (trap state $|t\rangle$ = electron is trapped at the center) and ground state (state $|v\rangle$ = neutral center, no free charges). The energy barriers involved in charge capture ($\Delta E_1$) and recombination ($\Delta E_2$) depend on the coupling strength between the trap and the lattice vibrations and on the energetic position of the trap level inside the band gap. For efficient non-radiative recombination both $\Delta E_1$ and $\Delta E_2$ should be as small as possible which means that the trap level should be close to the middle of the band gap

We are in favor of the idea of a complex because some experimental observations indicate that fluctuating quenchers are formed in specific locations rather than anywhere in the crystal[27,29]. One of these observations is the reproducibility of trends of the average PL intensity and PL blinking for consecutive cooling and heating cycles (Figs. 1g–j, 2d, and 4). It implies that the quenchers must keep their individuality after each switching from active states to the passive and back. Thus, they cannot be just transient like, for example, iodide interstitials or vacancies, which form, diffuse in space and then annihilate in arbitrary regions of the crystal. This is in line with previous experiments employing super-resolution microscopy, where it was observed that the position of the quencher does change beyond the detection limit of a few tens of nm over consecutive activation/ deactivation cycles spanning over several minutes[23]. Our argumentation does not exclude the involvement of the above-mentioned iodide-related defects in the quenching process However, based on our observations we stress that the fluctuating quenchers are rather fixed in space, in a particular local environment, where they keep their general properties over time. This should be considered in future models of non-radiative centers. Localization of the quenchers in different environments could also explain why the activation barriers for switching observed here are distributed over a very broad range. A scheme of a localized and a mobile defect forming a quenching complex, e.g. a donor-acceptor-pair is illustrated in Fig. 7a.

In principle, not only the intrinsic defects, but also residual solvent molecules, water, oxygen or structural defects can either directly be part of such complexes or indirectly influence them. In any of such cases, migration of at least one of the species could lead to the switching between a passive and an active configuration.

For further theoretical support of the hypothesis of quenching complexes, it would be desirable to carry out studies of different combinations of defects, which also are not limited to the intrinsic compounds of the material. Moreover, it would be helpful to address the aspect of metastability explicitly in computational studies, in order to learn which process (i.e. dissociation of complexes, change of the oxidation state of e.g. iodine or its vacancy or something else) could explain the observed switching dynamics on the very broad time scale.

Note that the idea that point defects individually are rather harmless while some of their complexes are strong non-radiative recombination centers nicely fits to the view on MHP as defect tolerant materials. Indeed, the concentration of simple defects can be very large as expected for a low temperature processed crystal, while the concentration of harmful defects can be still very low (e.g. $10^{10}$–$10^{12}$ cm$^{-3}$)[59] since they need two or more species to come together in order to form.

The decrease of $\kappa(T)$ upon cooling can come from a higher radiative rate, a lower non-radiative rate per quencher or a combination of both factors. More efficient radiative decay could result from a higher fraction of excitons, when the quenchers reveal different capture cross sections for excitons and charges[18]. Based on the Saha-Langmuir equation, we estimated the ratio between excitons and free charges for the studied nanocrystals under the experimental conditions applied in this work (see Supplementary Note 6). It turns out that, at room temperature, the fraction of free charges is relatively large, most likely above 90%. At temperatures as low as 80 K on the other hand, an excitonic fraction on the order of 50% is possible. However, for the blinking amplitudes, we have observed a decrease by a factor of 10, when the sample was cooled from room temperature to 80 K. Thus, the estimated reduction of the free carrier population is less pronounced than the observed reduction of the quenching efficiency and we conclude that there must be other mechanisms

that limit the quenching efficiency at low temperatures. A more likely reason is the temperature-dependent bimolecular rate coefficient, which was reported to increase by more than one order of magnitude when MAPbI$_3$ is cooled from room temperature to 100 K[60].

Additionally, the non-radiative rate can become lower at low temperatures due to the presence of thermal barriers in the recombination process. Thermal barriers should arise during (1) carrier diffusion toward the quencher; (2) trapping, and (3) recombination (see Fig. 7b). Since we do not observe trap filling (see Supplementary Note 4), we assume that recombination (annihilation of an already trapped carrier with a diffusing carrier) is not the rate-limiting step. Hence, suppression of diffusion and/or initial trapping remain as potentially limiting factors on the quenching efficiency at low temperatures.

Partial localization of excitations (charge carriers or excitons) in so-called band tail states is a common phenomenon in semiconductors with a disordered lattice potential[21] including perovskites[61,62]. This phenomenon becomes particularly pronounced at low temperature, which could make the quenching process diffusion-limited. In this case only a restricted volume around the spatial location of the quencher can be quenched efficiently. A higher temperature, on the other hand, elevates the excitations from the localized states above the so-called mobility edge, making it possible to reach the quenchers present in the crystal. This model is often used to explain temperature induced quenching in semiconductors[21,61]. By investigating the correlation between the PL lifetime and the intensity over the course of blinking[63,64], we find that activation of the quencher leads to both static (lifetime does not change upon quenching) and dynamic (lifetime decreases proportionally) quenching processes. This means that quenching of a fraction of the photogenerated species could indeed be limited by diffusion (Supplementary Note 7) and thus, the quenching volume would decrease with temperature. Note that in larger-sized MAPbI$_3$ crystals, diffusion-limited quenching has also been observed directly at room temperature[23].

In addition, it is known that the trapping process on a deep state itself is limited by energetic barriers[65–67]. The theory of this multi-phonon recombination in semiconductors was developed over many years starting with works of Huang and Rhys in 1950s[66,68] and has some analogy to Marcus theory of electron transfer in molecules[65,69]. Recombination occurs as a transition between potential energy surfaces of a system consisting of a local recombination center and a charge in a band (Fig. 7c). For charge trapping, the system must reach the crossing point of the potential surfaces, which requires energy $\Delta E_1$. The final recombination with the opposite charge also requires overcoming a barrier $\Delta E_2$.

In summary, we cannot make a conclusive statement about the underlying process that limits the quenching efficiency in the regime of low temperatures. Both, an increasing efficiency of radiative decay and hindered non-radiative processes could explain the observations and these scenarios are not mutually exclusive.

Although our Monte-Carlo simulations can qualitatively mimic the main experimental observations, both the model and the simulation have some limitations. Regarding the switching, one problem is that the initial population of either active or passive states is determined in the simulations randomly, which is justified only at thermal equilibrium. However, at low temperature, the switching rates can become very slow compared to the measurement time and the quenchers should freeze in one of the states without changing upon further cooling. The observation of hysteresis in the temperature dependent PL intensity and switching rates in some of the crystals (see e.g. crystals 175, 190, and 193 in Supplementary Figures 2 and 5) indicates that this

phenomenon is actually present in the investigated crystals, even though it is currently not considered in the model. Moreover, in some cases, the experimentally observed temperature dependence of the switching rate is much stronger than predicted by the model (see e.g. Figure 2d). This important experimental observation indicates that jumping between active and passive states might be coupled to the population of a particular phonon mode of the crystal or vibration of the local center. In this case assuming a temperature-independent attempt rate $k_0$ is an oversimplification. Consideration of such aspects in future studies will help to gain further insight into the nature of the luminescence quenchers in metal halide perovskites.

From the discussion above it becomes obvious that the PL enhancement upon cooling is most likely caused by a concerted contribution of many factors. Therefore, explaining PL enhancement by an increase of the concentration of emitting excitons vs. dark free charges, as suggested previously[15], and relating the Arrhenius activation energy obtained from fitting $PL$ $(T)$ to the exciton binding energy is a great oversimplification. The relation of the PL enhancement to the properties and the nature of the PL quenching defects explains why fitting of $PL(T)$ for different samples yields greatly varying activation parameters $\varepsilon$ ranging from 20 to 70 meV[14,15,19,20] and why we observed remarkable differences from one crystal to another in the slope of $PL(T)$ (Fig. 1g–j and Supplementary Figure 2). These findings help to understand the slope of $PL(T)$, which is of particular relevance when it serves as observable in device optimization procedures. Moreover, the analysis of temperature dependent PL blinking established in this work offers a novel approach to gain fundamental insights into non-radiative processes in OMHPs at the level of individual quenchers. There is a growing number of publications on blinking in perovskites demonstrating that this phenomenon is present in a variety of samples including micrometer-sized crystals[23], samples prepared in different atmospheres[26,36] and even perovskite thin films[70]. Based on this evidence we stress that blinking seems to be an intrinsic feature of metal halide perovskites. Thus, even though there might be quantitative differences in their formation depending on the preparation route, the fluctuating quenchers observed in this work are a general phenomenon that has to be considered for further improvement of the material. We have demonstrated that fluctuating PL quenchers contribute significantly to non-radiative decay, but their switching indicates that they can in principle be passivated. Stabilizing the passive state of these quenchers (e.g. chemically) thus appears to be a promising strategy to increase the material performances.

In conclusion, we point out that according to our findings non-radiative recombination in MAPbI$_3$ is partially caused by non-radiative channels which undergo random fluctuations between active and passive states. These fluctuations become apparent as PL blinking in 70 nm sized nanocrystals and they are caused by about 3 to 5 quenchers per crystal. Upon cooling the crystals down from 300 to 77 K, the time-averaged PL intensity increases by 1 to 2 orders of magnitude and PL blinking greatly reduces for temperatures below 200 K. Both the observed temperature dependent PL intensity and the blinking dynamics are highly individual from crystal to crystal and often fully repeatable in consecutive cooling-heating cycles. For many crystals the dependence cannot be approximated by the Arrhenius law. We propose that this peculiar behavior comes from the presence of several non-radiative centers (quenchers) per crystal having potential barriers between active and passive states. Based on this idea, we describe the switching using a simple model, from which we estimate that activation energies of the switching of individual quenchers must be broadly distributed from 0.2 to 0.8 eV. This range matches the range of reported energy barriers for ion

migration in perovskites. Therefore, it is likely that the random switching is caused by diffusing ions which can passivate or activate a non-radiative center, whose energetic position is determined by the local environment. We propose that the apparent temperature dependence of the PL yield comes from a concerted effect of hindered excitation diffusion, slowing down of trapping in fluctuating quenchers and decreasing non-radiative recombination via the constant quenching channel (presumably the surface) upon cooling. Therefore, it can be only indirectly related to the exciton binding energy. The success of our approach indicates to the community an avenue towards experimental insight to charge recombination at the microscopic level, and comprehending loss mechanisms in nano-structured semiconductors in general.

## Methods

**Sample preparation**. The CH$_3$NH$_3$PbI$_3$ (MAPbI$_3$, MA = CH$_3$NH$_3^+$) precursor solution used in this experiment was prepared by a one-step method. 461 mg PbI$_2$ and 159 mg MAI were dissolved in 1.25 mL of a mixture of the solvents dimethylformamide (DMF) and dimethyl sulfoxide (DMSO) with a ratio DMF: DMSO = 4:1. The solution was heated to 60 °C under stirring for 2 h. In order to obtain single MAPbI$_3$ nanocrystals, the stock solution was diluted to 1% and dropcasted on a silicon substrate in ambient atmosphere. Then the sample was covered with a glass plate and annealed at 90 °C for 30 min.

**PL microscopy**. For excitation of the crystals, we used the 458 nm continuous wave output of an argon laser. A set of neutral optical density filters in the excitation path allowed us to vary the excitation power over several orders of magnitude. The excitation fluence in the temperature dependent measurements was 0.3 ± 0.2 W cm$^{-2}$, depending on the position of the crystals in the excitation profile of the beam (see Supplementary Note 2). Precise knowledge of the local excitation power density is needed for a meaningful comparison of the crystal intensities and intensity jumps in the blinking transients. The samples were mounted in a continuous-flow cold finger microscopy cryostat (Janis, ST 500), allowing us to control the sample temperature between room temperature and 77 K. Between two measurements at different temperatures, the laser illumination was blocked. After reaching the temperature for a new measurement, we waited for 5 min to ascertain that the temperature had stabilized.

Measurements with a spatial resolution of ca. 0.6 μm were carried out with an inverted wide-field fluorescence microscope based on Olympus IX-71 frame and the PL images were recorded with an EM CCD camera (Princeton, ProEM 512 B). The image acquisition time was 50 ms and we recorded blinking movies over 100 s, yielding 2000 frames per movie. Note that these settings also set limitations to the switching times that we are able to detect. Our selection of the integration time represents a compromise between time resolution of the experiment and signal quality. For the power sweep (see Supplementary Note 4), the acquisition time was 100 ms and the overall duration of the measurement 50 s. In order to check our results for consistency and for a potential influence of temperature hysteresis, we performed the same measurements twice for cooling and heating of the sample under careful control of the sample position in the excitation spot.

By inserting a narrow slit and a diffraction grating into the detection path of the microscope, the setup could also be used to measure emission spectra. In order to demonstrate that the optical properties of the perovskite nanocrystals investigated here are in line with other literature reports, we recorded absorption and emission spectra, as detailed in Supplementary Figures 10 and 11.

**Blinking data analysis**. For further analysis, the recorded movies were corrected for sample drift (usually smaller than 2 pixels, corresponding to 400 nm). In a next step, we corrected the recorded movies with respect to each other, such that the investigated crystals were located at the same coordinates for each temperature. In order to obtain a large number of crystals for analysis, we applied a localization algorithm to the time-averaged data recorded at 140 K. Overall, a number of 197 crystals was localized and for each temperature, we extracted the PL intensity traces at the respective coordinates. Additionally, we extracted the background noise in the vicinity of the crystals at the selected coordinates and the mean of this background was subtracted from the intensity traces for further analysis.

In order to distinguish blink events from noise-induced intensity fluctuations, we analyzed the extracted intensity traces in two steps: First, we applied a Bayesian inference method to detect both intensity change points and stable intensity levels[45]. The method tests if a certain segment of the intensity trace can be described by a single Gaussian model, which is then considered as segment of constant PL intensity, or if there is a change point. In the latter case, the data is better described by two distinct Gaussian distributions and the algorithm localizes the intensity switching event in the trace. However, very short blink events, where the residence time at a particular intensity level is 150 ms (three frames) or less,

cannot be detected by this statistical method. Thus, in a second step, we detected outliers from the intensity distributions of the assigned stable levels.

This procedure was carried out for the intensity traces of all crystals at all temperatures. However, in some cases the signal-noise-ratio was not sufficient to draw meaningful conclusions about the underlying blink events. In order to improve the reliability of the dataset, we therefore implemented a cleanup procedure, where the extracted model traces were normalized by the standard deviation of the corresponding background. This 'standardized' PL intensity is directly related to the underlying signal-noise ratio and allows to define a threshold intensity, below which the data is not considered for blinking analysis. For the quantitative analysis (Fig. 3), we determined a minimum mean standardized intensity of 2, below which data was not considered. Furthermore, the size of the investigated crystals is very small, thus we assume that the PL intensity profile must look similar to the point spread function of the setup. Otherwise, it is rather likely that several crystals contribute to the intensity trace extracted at a specific position. Thus, in a second cleanup step, conformity with the point spread function was controlled by a two-dimensional Gaussian fit (for details of the selection procedure see Supplementary Note 1).

**Simulation details**. Based on Eqs. 2 and 3, we carried out Monte-Carlo simulations to model blinking transients. In order to demonstrate that our model captures the main experimental observations, we tried to reproduce the temperature dependent intensity traces of several crystals by adjusting the energies $E_{p\to a}$ and $E_{a\to p}$ for a number of 4 or 5 quenchers per crystal (see Fig. 6 and Supplementary Figures 6–8). From these energies, we determined the parameters $\tau_{active}$ and $\tau_{passive}$, which are used to define exponential probability distributions, from which the random switching times are obtained. These random times enter the model as sequences of zeros and ones contained in $Q_i(T,t)$. For each temperature, we determine the initial state of a quencher (either active or passive) by calculating the probability to find the quencher in the passive state:

$$P_{passive} = \frac{\tau_{passive}}{\tau_{active} + \tau_{passive}} \tag{5}$$

The initial state is then randomly determined with a probability $P_{passive}$ to be passive and $P_{active} = 1 - P_{passive}$ to start in the active state.

We set the time step for the simulation to 0.1 ms, well below the experimental integration time of 50 ms per frame. After generation, the simulated transients were averaged over intervals of 50 ms to mimic the experimental conditions. The range of switching rates, which can be experimentally observed as blinking, is limited by the measurement time (lower limit, 100 s) and the integration time (upper limit, 50 ms). In order to save computing time, we did not compute the switching transients of quenchers if either $k_{p\to a}$ or $k_{a\to p}$ was above $(0.3\,ms)^{-1}$, because the intensity fluctuations induced by these quenchers would be averaged out in the experiment. Instead, we considered the time-averaged influence of these quenchers. Details of this procedure are described in Supplementary Method 2.

## Data availability

The data that support the findings of this study are available from the corresponding authors upon reasonable request.

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

## Acknowledgements

The work was supported by the Swedish Research Council, project 2016-04433. J.L. thanks the China Scholarship Council (CSC No:201608530162) for a PhD scholarship. A.D. acknowledges the Carl Trygger foundation and M.G. acknowledges the Wenner-Gren foundation for a postdoctoral fellowship (UPD2017-0223). J.H. acknowledges financial support from the Research Foundation-Flanders (FWO, Grant Nos. G.0962.13, G.0B39.15, G.0197.11, G098319N and ZW15 09 GOH6316), KU Leuven Research Fund (C14/15/053), the Flemish government through long term structural funding Methusalem (CASAS2, Meth/15/04), the Hercules foundation (HER/11/14), and the Belgian Federal Science Policy BELSPO (IAPVII/05). B.L. and R.C. thank the Research Foundation-Flanders (FWO) for a doctoral and postdoctoral fellowship, respectively. A.M acknowledges funding from the German Ministry of Education and Research (BMBF) within the program "NanoMatFutur" (Grant No. 03XP0091).

## Author contributions

J.L. and B.L. prepared the samples for characterization. Under supervision of I.G.S. and J.H., M.G., B.L., A.D., A.K. and A.M. planned and carried out the PL experiments. A.D. carried out SEM measurements. M.G., B.L. and R.C. analyzed the PL data. R.C. wrote an algorithm to distinguish blink events from noise in the recorded blinking transients under supervision of J.H., I.G.S. and M.G. developed the switching model and M.G. carried out Monte Carlo simulations of PL blinking. M.G. and I.G.S. wrote the manuscript. All authors were involved in discussions and the writing process.

## Additional information

**Competing interests:** The authors declare no competing interests.

