## [Peer Review File · Nature Communications]

Reviewers' comments:

Reviewer #1 (Remarks to the Author):

Gerhard et al presented a comprehensive study of PL blinking in single MAPbI₃ nanocrystal, at various temperatures. The PL blinking is ascribed to random switching of about 3-5 quenchers in 50 nm sized nanocrystals between active and passive states. Upon cooling the crystals, the time-averaged PL intensity increased by 1-2 orders of magnitude and PL blinking greatly reduced. They confirmed both the temperature dependent PL intensity and also the blinking dynamics were highly individual. Based on thermal-activation mechanism, they propose a model to describe the observed reduction of blinking upon cooling and determine energetic barriers of 0.2-0.8 eV for enabling the switching process. Nonradiative mechanisms has been intensively investigated in perovskites due to intimately correlated with the photovoltaic and photonic applications. PL blinking has been studied in perovskite large grains/nanocrystals that are different from the conventional semiconductor QDs (including perovskite QDs). However, the intrinsic physical mechanism to date still reminds poorly understanding.

This manuscript provide comprehensive observation and analysis, representing a significant progress with respect to the previous reports. It therefore provide a valuable reference for perovskite community. I recommend publishing this manuscript if authors can address the following concerns:

1. For considering the detailed physical mechanism, authors should estimate excitation carrier density according to experiment details, such, as shape and size of nanocrystals, objective and beam size.
2. "it is likely that the random switching is caused by diffusing ions". It is estimated that there are around 3-5 quenchers in 50 nm nanocrystal. The 50 nm nanocrystals may be simplified as multiple QDs that emit/quench synchronously. Is it possible to estimate effective quenching volume? correlating with screen effect?
3. Photon lifetime statistics at different temperatures can further support the conclusion.
4. Given the higher thermal energy at room temperature and thus highly active quencher, 50 ms integrating time is too long. Some shorter time blinking may be helpful.
5. Basically, excitation is also illumination that can effectively activate ions thus switch on the quencher. Can authors provide some excitation intensity dependent PL blinking?

Reviewer #2 (Remarks to the Author):

This manuscript presents a detailed experimental study of photoluminescence intermittency in organo-metal halide perovskite nanocrystals, accompanied by an extensive theoretical discussion, modeling and analysis.

Given how important perovskites are, and, given that the detailed microscopic mechanism of fluorescence/luminescence intermittency is still not understood, I am inclined to agree that this work is important and relevant to the wide audience served by Nature Communications.

However, considerable improvements/corrections/clarrifications should be made on the manuscript before it is ready (at least in my opinion) for publication. I am listing my comments and questions in the (approximate) order in which they appear in the manuscript.

* on p. 2 the authors state that the extraction of quencher concentration from PL trajectories was "hitherto impossible". While this may be new for the perovskite nanocrystals, a map of the nonradiative recombination centers, and its time evolution has been computed from fluorescence images for reduced graphene oxide (see, for example Nano Lett 15, Pages: 4317-4321 (2015)).

* it is true that the strong temperature dependence documented by the authors in this manuscript is in sharp contrast to the virtually temperature independent blinking seen in colloidal quantum dots, rods, wires, etc. Therefore I agree with the (key) statement on p. 8 that the mechanism of blinking must be different in perovskites than in quantum dots.

* it is true, as the authors state on p 4, that just with a few quenchers are enough to explain most PL trajectories. However, it is not entirely true that quantum dot blinking occurs between two defined levels. Typically, there is a wide distribution of states that contribute to the histogram of states participating in blinking, and _sometimes_ the histogram has two maxima, but there are well known cases of three levels showing up up (see Mulvaney's data).

* on p. 9, "the increase of $K(T)$ upon cooling ... " should in fact be "decrease"

* the diffusion-limited quenching mechanism described on p. 9 is qualitative at best, seems more of a speculation, unless the authors can show the emergence of a diffusion-related time or length scale.

* the freezing mechanism (when a quencher gets stuck in a certain state) mentioned on p. 10 is very interesting. What is the freezing temperature? What are the phonon modes that the authors suspect are coupled to the active-to-passive transition of the nonradiative recombination centers?

Further comments, questions more specific to the experimental methods and materials preparation:

It would first be nice to know more about the material system being studied. We know that these are MAPbI₃ nanocrystals. Not explicitly stated are the following:

What are their average sizes and what is the corresponding size distribution within an ensemble? Very late in the text a number such as 50 nm appears. Is this the average size of the particles being studied?

What does the ensemble absorption and emission profile look like?

What is the ensemble emission quantum yield?

Regarding the images in Figure 1. How do the authors know that one is looking at individual nanocrystals and not small aggregates below the diffraction limit? Are there corresponding AFM measurements or SEM measurements of studied samples? Alternatively, are there antibunching measurements –assuming that they are quantum emitters?

How does the data in Figure 1 panel e change with total movie duration. If you analyze the data for 5 seconds total, 10 seconds total or 30 seconds total does the Arrhenius fit results change significantly? It would seem to the reviewer that there could be duration dependence of the overall ensemble intensity depending on the blinking statistics of individual emitters. For example, if you had exponential or power law distributed blinking, the longer one samples the greater the likelihood of encountering rare events.

What are the circled points in Figure 1, panel e? This should be stated in the figure caption.

Similarly, what is the excitation wavelength and excitation intensity for the data in Figure 1. Perhaps

this should be stated in the figure caption or at least some mention of it made in the accompanying text.

For the data in Figure 1 g-j, the reviewer is curious if there are accompanying spectra that show redshifts with decreasing temperature. Are the spectral energies of the emitter consistent on the cooling and heating paths? Part of the reason for the curiosity is that these spectral shifts would provide additional evidence for cooling and heating and if calibrated using say an ensemble measurement would provide local single emitter temperature estimates –as opposed to using the temperature controller suggested temperature. Would the Arrhenius behavior deviations then rectify themselves or become worse?

What is the excitation wavelength and excitation energy for the data in Figure 2? Again, this information should be stated nearby. The one thing the reviewer would like the authors to include in this discussion is the average number of excitations in the particle. Presumably, one wants to conduct blinking type measurements in a regime where Auger processes do not dominate. State also the assumed Auger coefficient. Additionally, temperature-dependent measurements should cause the absorption to shift. How much is the absorption coefficient changing between temperature and how does that impact the effective excitation densities in the nanocrystals?

How do the blinking trajectories change with excitation intensity below the onset of Auger? Does blinking frequency increase with increasing excitation intensity as might be expected? Conversely, does going to very low intensities increase the duration of on intervals at a given temperature?

How is the nanocrystal spectrum evolving during a given trajectory? Is the emission energy stable or is it changing with time indicating a spectral diffusion process linked to the intermittency?

Do the individual nanocrystals eventually photobleach?

What is the estimated emission quantum yield of an individual nanocrystal during a given on period at different temperatures? This can be estimated knowing the temperature dependent absorption coefficient of the particle and the estimated collection efficiency of the instrument. Is unity quantum yield ever achieved? A corollary question would be what the lifetime traces looked like during on and off intervals. Presumably during stable on portions of a trajectory one would see an exponential decay which together with the quantum yield estimate would enable one to suggest what radiative and non-radiative rate constants were involved? Is this data available as it would help.

For temperature dependent measurements, what is the exciton versus free carrier fraction in the nanocrystals, assuming that these crystals are outside their confinement regime, that dielectric contrast does not enhance bulk binding energies and also employing the Saha equation. Next, there are presumably a larger fraction of excitons present at low temperature while relatively few at room temperature. Are the authors suggesting that the quencher sites do not distinguish between free carriers and excitons? If not, are the authors suggesting subcategories of quenchers that should be introduced into subsequent kinetic modeling?

The labeling in Figure 4 must be improved. It is not clear which trace is Figure 4b. It was also unclear to the reviewer how the switching rate was estimated for this Figure. Perhaps some mention of this would improve things.

In equation 2, it was not clear why physically each quencher needed to have the same decay rate constant.

The authors should probably define E_a , E_x and E_p close to Figure 3 where these parameters are introduced.

The reviewer thought that the ion migration hypothesis was interesting but did not understand it. Perhaps this should be explained in some more detail so as to establish within the context of the hypothesis what an active defect configuration looks like versus a passive one. Along the same lines and as suggested by the authors -why would ion migration necessarily be reversible in the sense of cooling and heating trajectories having the same quencher configuration.

Next, carrier mobilities are quite large in hybrid perovskites. It would seem that the physical act of a carrier diffusing to a local quencher and extinguishing the excitation would be very efficient. Perhaps too efficient. The timescales of carrier diffusion to a site within a 50 nm distance must be extremely short. Why would sustained on times be even possible with any regularity given multiple quenchers per nanocrystal?

It was not clear how the early ensemble 63 meV activation energy linked up to the 0.2-0.8 eV single quencher barriers suggested from subsequent temperature-dependent measurements.

The authors say that they adjusted the parameters of the simulation (such as the number of quenching sites to ~1-3 per nanocrystal) to improve the agreement with the experiment. What features of the time trajectories were improved from this? It appears the main feature is the intensity histogram. Did time-correlation features such as on and off-time distributions or power spectral density improve?

Reviewer #3 (Remarks to the Author):

In this manuscript, Gerhard et al. report on the non-radiative decay in perovskite nanoparticles (NPs) and explain the underlying mechanism based on the temperature dependent photoluminescence (PL) blinking experiment. In brief, the authors attribute the blinking behavior to the existence of individual non-radiative recombination centers, which is associated with the ion migration. Overall, the topic is interesting and important; however, the model is simplified without considering the interaction between photogenerated charges and ions as well as other important factors. Therefore, the referee cannot recommend its publication before the authors fully address the following questions.

1. The title is "Microscopic insight into non-radiative decay in perovskite semiconductors from temperature-dependent luminescence blinking". However, in this paper, the system under investigation is based on perovskite NPs. It is well known that during the formation of NPs, there exists a large amount defects and surfaces, which make them entirely different from the property of periodical structure crystals. It might be difficult to justify that the same mechanism is still valid in perovskite single crystal or even perovskite thin film.

2. In the sample preparation part (Page 11), the authors claim that the samples were fabricated in ambient atmosphere. It is widely accepted that the humidity and oxygen molecules have significant impact on the performance of perovskite device, especially in the perovskite NPs, which possess large surface area to volume ratio. Is there any difference between the device fabricated in N₂ filled glovebox and ambient condition?

3. I wonder whether the authors have observed the similar blinking behavior in compact perovskite thin film.

4. According to Galland et al. [Nature 479, 203(2011)], it is known that there are two kinds of PL blinking, (1) conventional one due to charging and discharging, (2) blinking due to charge fluctuation in the sites. By investigating the PL intensity against the lifetime, it is possible to distinguish these two types of blinking mechanism. I wonder whether it is possible to present this result.

5. The excitation intensity in the experiment is 0.3 ± 0.2 W/cm². This is a very high intensity which might lead to the increased temperature of the sample surface. Please comment on the possible influence.

6. The authors attribute the blinking behavior to the ions as the quencher. It is important to consider the influence of photogenerated charges under light illumination, as well as the interaction between the photogenerated charge carrier (electrons and holes) and ions (defects).

7. Figure 1(g) to (h) depict the temperature dependent PL. However, in the method part, the authors did not provide any experiment details, such as the heating speed, cooling speed, in dark or continuing illumination, etc. In addition, Figure 1(g) to (h) shows negligible hysteresis during the heating and cooling process. However, in Fig S7, cr 193, cr 190 and cr 175 exhibit significant hysteresis behavior. Please give the comment on the behavior.

8. On page 8, the authors indicate that the activation energy is between 0.2 and 0.8 eV. Additionally, the authors mention that 300 K correspond to 0.025 eV. Hence, it is nearly impossible to overcome the barrier only by the room temperature. In addition, the authors said they would provide detailed discussion on section 10, but I did not find the discussion in SI. Please provide detailed information and discussion.

9. From Page 17 to Page 18, I cannot find Figure 7 to Figure 12 in the main text.

10. In Figure 6(b), E1 and E2 should be $\Delta E1$ and $\Delta E2$.

11. The authors should be careful with the Reference part. Ref (8), (25), (42), (54), (58), (59), (61) are not properly cited.

Response letter

First of all, we would like to thank all reviewers for their helpful and detailed remarks, close reading and general positive attitude to our work. Based on their suggestions, we carried out further experiments to study how the PL intensity in a blinking transient is related to the PL dynamics. We also performed a more detailed analysis to determine the crystal size and excitation density. Our efforts led to changes in the main text, they appear as additional chapters in the SI and we also attached a supplemental material for review only where some of the concerns of the reviewers were addressed. Important changes in the text and SI are marked by blue color. We believe that our additional work strengthens the major statements made in the paper and hope that the revised version can be further considered for publication.

Reviewer #1

1.1. For considering the detailed physical mechanism, authors should estimate excitation carrier density according to experiment details, such, as shape and size of nanocrystals, objective and beam size.

We thank the reviewer for this suggestion. In the revised version, as a first step, we determined the crystal size and size distribution in a more accurate way by performing a quantitative analysis of SEM data recorded for two different regions of the sample. On average, the crystal diameter is about 70 nm. Taking into account the profile of the excitation beam, this allowed us to estimate the average number of excitations per crystal. Considering the absorption coefficient and estimated lifetime of the carriers (< 50 ns), we estimate an average number of one excitation per crystal and time.

Implemented changes:

Details of this analysis and all relevant parameters are given in **SI sections 2 and 3**. In the main text, we now mention the excitation power density in the caption of **Figure 1**. The crystal diameter is mentioned in the caption of **Figure 2** and in addition we corrected our previous estimate of 50 nm in the first paragraph of the discussion on **page 10** to 70 nm.

1.2. "it is likely that the random switching is caused by diffusing ions". It is estimated that there are around 3-5 quenchers in 50 nm nanocrystal. The 50 nm nanocrystals may be simplified as multiple QDs that emit/quench synchronously. Is it possible to estimate effective quenching volume? correlating with screen effect?

Indeed, it is maybe possible to see a large crystal as a collection of small volumes each when each of them is treated as independent QD. However, we do not feel that such approach is really useful for us. In any case, synchronous quenching must be due to a mechanism of some "communication" between those QDs which is just excitation migration. As soon as "communication" is really efficient we cannot treat these "QDs" as individual objects anymore. Since the PL spectrum of our crystals is very close to that of bulk material, we think that there are no small volumes in the crystals which need to be seen as objects with quantum confinements and all other features intrinsic to real QDs.

However, we definitely can estimate the quenching volume from our data. In some particular cases, we performed both SEM and blinking measurements on the same objects. Two examples are presented in the **SI, Figure S3**. From the blinking transients, we determined the fraction of the overall intensity that can be quenched by a single quencher. The estimated quenching volume then corresponds to this fraction of the crystal volume; it is about 70 % of the whole volume for the selected crystals, assuming that the whole crystal is emitting. This corresponds to a quenching volume on the order of 10^5 nm^3 .

Note that if the diffusion length of the excitations is larger than the size of the crystal, then at the low excitation limit (no saturation of the quencher) the quenching volume is simply limited by the volume of the crystal. We believe that this situation applies to our case at least at room temperature (see responses to later comments). So, since our intensity-lifetime correlation suggests that diffusion does not set a major limitation to the quenching in most cases, we hypothesize that the actual quenching volume is larger (see **section 14** of the SI for detailed discussion).

Implemented changes:

We added the calculation of the quenching volume to the **SI, section 2**.

1.3. Photon lifetime statistics at different temperatures can further support the conclusion.

We thank the reviewer for this suggestion and indeed detailed lifetime measurements can help. In response to the reviewer's suggestion here and some later comments, we carried out time-correlated single photon counting measurements of some selected crystals under variation of the sample temperature. Operating the system in the time-tagged mode allowed us to investigate the PL decay for different intensity levels of the recorded blinking transient. The data and a detailed discussion are presented in the **SI, section 14**.

When the lifetimes are correlated with the PL intensities measured in the blinking transient, there are two limiting cases: In the first case, the PL intensity is proportional to the PL decay time. When this is the case, radiative and non-radiative recombination deplete the same pool of excited states and the amount of non-radiative decay is only limited by the capture cross section of the quencher. In the second case, the lifetimes do not change with the PL intensity. Such behavior is expected, when the non-radiative process is spatially constrained, which means that the crystal reveals extended regions, from where charges do not diffuse (or do not have enough time to do so) into the quenching sites.

Here, we find evidence for a combination of both cross-section (rate limited) and diffusion limited luminescence quenching. Based on this finding we can speculate that the quenching volume could decrease with decreasing temperature, when diffusion towards the quenchers at low temperatures becomes more difficult. However, the presently recorded data is not of sufficient quality and quantity to allow further conclusions concerning the temperature dependent diffusion of the luminescent species. Note that the crystals investigated here are relatively small, whereas in bigger crystals, diffusion-limited quenching has been observed even at room temperature (see Merdasa et al., *ACS Nano* 2017, 11, 5391).

Implemented changes:

As indicated by several of the reviewers' comments, we believe that the lifetime-intensity correlations are of broader interest and thus represent a valuable supplement to our work. We thus added the analysis and discussion performed here to the **SI (section 14)**. In the main text, we refer to this issue on **page 12**, in connection with a discussion of the question if quenching could be limited by diffusion.

1.4. Given the higher thermal energy at room temperature and thus highly active quencher, 50 ms integrating time is too long. Some shorter time blinking may be helpful.

We agree with the reviewer that time resolution limits the speed of the processes we can observe. The integration time of 50 ms indeed does not allow us to detect very fast switching dynamics, which we expect to occur. We chose an integration time of 50 ms mainly due to practical aspects. In principle, faster dynamics could be detected with the CCD camera, but that would require a reduction of the detection area. Thus, less data would be available for quantitative analysis. Moreover, there is a limitation on the number of images we can save, that is why, decreasing the exposure time automatically decrease the total time observation window which will not allow to see long switching events. We are currently working on solving these technical difficulties in our setup, and we hope to use much longer transients in our future research.

Moreover, a shorter integration time lowers the signal-to-noise ratio. Even though shorter blink events can be detected with a shorter integration time, it becomes more difficult to distinguish them from noise-induced intensity fluctuations. Note that the detection efficiency of our setup is rather low, because no oil immersion objective could be used due to the cryostat, in which the sample was mounted. As an example, we investigate a blinking transient that was recorded with the APD with high temporal resolution in **section 1** in the SM for review. The intensity transient is plotted with several binning times. Although the essential features can be resolved with an integration time of 50 ms, more details become apparent for shorter binning. However, these features are superimposed with noise, so a compromise between signal quality and time resolution has to be found.

In summary we believe that a higher time resolution would not essentially change the experimentally observed features, such as the temperature dependencies found for the switching rates and blinking amplitudes. Even for a shorter integration time we expect that a remarkable fraction of the switching could not be resolved. This aspect, however, is considered in our modeling, which takes the integration time into account (**see section 9 in the SI**).

Implemented changes:

Regarding the manuscript, we feel that we should point out to the reader more clearly that the detectable range of blinking times is limited. We added a corresponding statement to the method section on **page 16**. In addition, the experimentally accessible range of switching rates is highlighted in the new **Figure 5**, which we inserted in the course revising the model section (for details see response to comment 3.8).

1.5. Basically, excitation is also illumination that can effectively activate ions thus switch on the quencher. Can authors provide some excitation intensity dependent PL blinking?

In order to investigate potential effects of the illumination intensity, we fabricated another sample following the same method as described in the manuscript and performed excitation intensity dependent blinking measurements. The results of the excitation power dependent measurement are summarized in **section 7 of the SI**. Surprisingly, we do not observe any clear effect of the illumination intensity on the switching behavior. The reason for this could be that the regime of excitation densities we probe is actually too high to see the onset of photo-activation that would lead to increased blinking activity. A study on photo-activated ion migration reports significant changes of the ionic conductivity for excitation powers below 0.02 W/cm^2 with a tendency to saturate at higher excitation powers (see Zhao et al., *Light: Science & Appl.* 2016, 6, e16243). In our experiment, however, we applied excitation densities ranging between 0.02 W/cm^2 and 12 W/cm^2 . We therefore speculate that the effect of photoactivation plays only a minor role.

Contrary to our results, there are also two studies of blinking in MAPbI_3 that demonstrated an effect of the excitation power on blinking (Yuan et al., *ACS Omega* 2016, 1, 148 and Tian et al., *Nano Lett.* 2015, 15, 1603). We suspect that our results differ from the former, because the range of excitation densities was again lower than the range applied in our work (0.001 to 0.16 W/cm^2 in the paper of Yuan et al.). In the latter, the excitation power dependence was associated with trap saturation, which is, however, also absent in the crystals we investigated. This we verified by analyzing the blinking amplitudes, which showed no saturation at different excitation intensities.

Implemented changes:

We included the updated version of the excitation power sweep in the SI of the manuscript (see **section 7**). There, we also added our investigation of the influence of photoactivation and some parts of the above-made discussion. In the main text, we briefly discuss the absence of photoactivation effects on **page 11**.

Reviewer #2

2.1. on p. 2 the authors state that the extraction of quencher concentration from PL trajectories was "hitherto impossible". While this may be new for the perovskite nanocrystals, a map of the nonradiative recombination centers, and its time evolution has been computed from fluorescence images for reduced graphene oxide (see, for example *Nano Lett* 15, Pages: 4317-4321 (2015)).

We are grateful to the reviewer for pointing out this very relevant publication which we were not aware of. In the mentioned study the authors used the idea which is also used in our work. Having extra spatial information one can disentangle the contribution of the concentration and strength of the non-radiative centers. In the mentioned work the spatial resolution was reached directly due to large lateral separation between the non-radiative centers in graphene (2D material). Here in perovskites we are dealing with 3D objects and much higher defect concentration, so, the spatial distance between quenchers is small. However, in our case it is the size of the crystal which gives us the spatial constraints.

Implemented changes:

We have modified the text in the introduction and cited the suggested work to emphasize that spatial mapping of fluctuating non-radiative channels has been shown in some systems. We reformulated the related text in the introduction with more focus on the temperature dependence, which is the actual novelty. The changes were made in the main text on **page 2**.

2.2. It is true that the strong temperature dependence documented by the authors in this manuscript is in sharp contrast to the virtually temperature independent blinking seen in colloidal quantum dots, rods, wires,

etc. Therefore I agree with the (key) statement on p. 8 that the mechanism of blinking must be different in perovskites than in quantum dots.

2.3. It is true, as the authors state on p 4, that just with a few quenchers are enough to explain most PL trajectories. However, it is not entirely true that quantum dot blinking occurs between two defined levels. Typically, there is a wide distribution of states that contribute to the histogram of states participating in blinking, and *_sometimes_* the histogram has two maxima, but there are well known cases of three levels showing up up (see Mulvaney's data).

We thank the reviewer for this comment. We were not considering this fact when citing the papers. The aim of this passage was to show that a threshold-based ON-OFF analysis, as often performed for quantum dot blinking, is not feasible in our case. In order to avoid the misleading statement that suggests that quantum dot blinking would only occur between two levels, we now give the references to the quantum dot blinking in a different context, in order to point out that in these particular cases a threshold-based analysis of “on” and “off” levels was feasible:

“... Due to the combined effect of several fluctuating quenchers per crystal, the blinking traces observed here show very complicated behavior. Thus, a conventional analysis of “on” and “off” times, as often performed for quantum dots,^{44,45} does not appear to be a good approach in our case. ...”

Here, we only cite the papers of Kuno et al. and Shimizu et al., who carried out a two-level-analysis. The other papers of Frantsuzov et al. and Efros & Rosen that were originally referred to here are now cited in passages where it appears more appropriate, i.e. in the context of the model of multiple recombination centers (**page 7**) and in the introduction where we refer to several blinking systems (**page 2**).

Implemented changes:

The changes in the text were made on **page 5** and the references on **page 2** were updated.

2.4. On p. 9, "the increase of K(T) upon cooling ..." should in fact be "decrease"

We thank the reviewer noting this mistake which is corrected now (**page 12** in the revised version).

2.5. The diffusion-limited quenching mechanism described on p. 9 is qualitative at best, seems more of a speculation, unless the authors can show the emergence of a diffusion-related time or length scale.

Indeed, it is difficult at the moment to assign a particular mechanism to the effect of the decreasing of the quenching efficiency. The discussion on page 9 is intended to summarize different scenarios that could lead to the observed reduction of quenching efficiency upon cooling. Diffusion is among these scenarios and, as we believe, it is not unlikely to occur, because carrier localization in band tail states at low temperatures is a well-known phenomenon in semiconductors and has also been reported for perovskites (see e.g. He et al., *Nat. Commun.* 2016, 7, 10896 and Wright et al., *Adv. Funct. Mater.* 2017, 27, 1700860). Such behavior does not contradict the fact that the intrinsic mobility increases upon cooling, because in luminescence we probe excitations that have thermally relaxed into local minima of the band edges, from where further transport may be prohibited. We also note that for blinking studies of bigger crystals, diffusion-limited quenching has indeed been observed, even at room temperature (see e. g. Merdasa et al., *ACS Nano* 2017, 11, 5391). Even though the crystals studied here are smaller than the diffraction limit of the optical microscope, our time-tagged blinking experiments (see **SI, section 14**) indicate that the efficiency of carrier migration towards the quencher may be limited. All these arguments support the hypothesis of limited carrier diffusion.

However, despite this evidence, we do not intend to propose an exclusive picture that is actually based on diffusion-limited transport towards the quenchers. As the reviewer pointed out, so far it is just an educated suggestion and more experiments are needed to give further support or disprove the idea. Other scenarios may apply as well to explain the reduction of the quenching efficiency upon cooling, but we would like to point out that we are actually discussing different scenarios in the text, in particular the changing ratio of radiative and non-radiative decay and the emergence of thermal barriers in the capture and recombination process.

Implemented changes:

To give further support for the diffusion hypothesis on **page 12-13**, we added our analysis of the intensity-lifetime-correlation to the supplemental material of the manuscript (**section 14**) and we refer to this analysis and to the paper of Merdasa et al. in the main text on **page 12**.

On the other hand, the text should not give the impression that the diffusion-hypothesis would be the most likely one. Therefore, we rephrased a sentence on **page 12** and added a paragraph on **page 12-13** where we state more clearly that the decrease of the quenching efficiency upon cooling can have different origins.

In order to improve the structure of this section, we subdivided the discussion of the quenching efficiency into several paragraphs, each of them referring to one of the scenarios we propose to explain the observations.

2.6. the freezing mechanism (when a quencher gets stuck in a certain state) mentioned on p. 10 is very interesting. What is the freezing temperature? What are the phonon modes that the authors suspect are coupled to the active-to-passive transition of the nonradiative recombination centers?

From the available experimental data, it is difficult to estimate a specific temperature, below which freezing occurs. However, for a few of the crystals that we investigated, hysteresis occurs in the PL intensity and switching rates when cooling and heating cycle are compared, as also pointed out by reviewer #3, comment 3.7 (see e.g. crystals 175, 190 and 193 presented in **sections 1 and 8 of the SI**). From this data, we get the impression that the temperature range of freezing phenomena is quite flexible, which might be related to the fact that we expect the activation barriers for switching to spread over a broad range.

The question, which particular phonons are driving the switching of the quenchers is very interesting. The fact that blinking is observed even for temperatures as low as 80 K indicates, that phonon modes with energies lower than 7 meV must be involved (kT at 80 K \approx 7 meV). According to Sendner et al., Mater. Horiz. 2016, 3, 613-620, the optical phonon energies of MAPbI₃ are 4 meV (TO), 8 meV (TO), 5 meV (LO) and 16 meV (LO), thus, beside the lower-energetic acoustic phonons, also optical phonons could be responsible for the observed blinking. The observation of more blink events at higher temperature could also be partially related to the activation of more phonon modes.

Implemented changes:

We mention the observation of hysteresis in the discussion section on **page 13**, related to the discussion on limitations of the model. We also changed the plots of the temperature dependent PL intensity in the **SI, section 1**. They now contain separate plots of the data from the heating and the cooling sweep to demonstrate that hysteresis behavior is also present in the PL intensity.

2.7. Further comments, questions more specific to the experimental methods and materials preparation:

a.) It would first be nice to know more about the material system being studied. We know that these are MAPbI₃ nanocrystals. Not explicitly stated are the following:

- What are their average sizes and what is the corresponding size distribution within an ensemble? Very late in the text a number such as 50 nm appears. Is this the average size of the particles being studied?

Generally, this material is very standard for the perovskite community, also the crystals are prepared by standard routine but using lower concentration of the precursor. This is how instead of a film we obtained individual crystals.

The number of 50 nm was estimated from SEM images of two individual crystals. Therefore, this value is not necessarily representative for the whole ensemble of crystals. We thank the reviewer for pointing out this inaccuracy. In order to give a more precise value for the crystal size, we performed a quantitative analysis of two larger groups of crystals based on their SEM images. Details of this analysis can be found in **section 2 of the SI**. From this, we estimate a crystal diameter of 70 nm. In addition, we show the size distribution of the studied group of crystals. Please, also see our detailed reply 1.1 to the Reviewer #1.

Implemented changes:

Details of this size analysis and all relevant parameters are given in **section 2 of the SI**. The crystal diameter is mentioned in the caption of **Figure 2** and in addition we corrected our previous estimate of 50 nm in the first paragraph of the discussion on **page 10** to 70 nm.

- What does the ensemble absorption and emission profile look like?

We agree that it is important to show the spectra for the readers. It is difficult to measure absorption spectrum of our samples due to the very low concentration of the material. In the revised SI we included now the absorption spectrum of a film (with not complete coverage though) prepared using the same route but higher concentration. This is the standard spectrum of MAPbI₃ semiconductor. We also include the PL spectrum of this sample at room temperature (see **section 15**).

For a few individual crystals, we recorded temperature dependent emission spectra, these are shown in **section 15 in the SI**. As typical for MAPbI₃, we observe a red-shift of the emission maximum with decreasing temperature, which is interrupted by an abrupt blue shift in the temperature range between 160 and 120 K due to the phase transition from tetragonal to the orthorhombic phase.

Implemented changes:

The new **section 15 in the SI** contains the recorded PL spectra and a brief discussion.

- What is the ensemble emission quantum yield?

The emission quantum yield (QY) in MAPbI₃ varies a lot depending on the preparation conditions. From the PL enhancement upon cooling of the ensemble-averaged emission by about two orders of magnitude we can estimate that the PLQY is less than 1 % at room temperature which makes sense due to the presence of surface quenching and no surface protection such as ligands, as typically used in QDs.

To obtain a more accurate picture, we estimated the QY of individual crystals counting the number of photons detected from them at given excitation conditions and knowing their size and collection efficiency of the setup. For the two crystals we obtain quantum yields of 0.2 % and 1.9 % at room temperature and values around 20 % at 77 K. Thus, despite the strong increase of the QY upon cooling, non-radiative channels are still present at 77 K.

Implemented changes:

Details of the estimation of the QY can be found in **section 4** of the SI. In the manuscript we refer to our estimate of the room temperature PL quantum yield on **page 3**.

- Regarding the images in Figure 1. How do the authors know that one is looking at individual nanocrystals and not small aggregates below the diffraction limit? Are there corresponding AFM measurements or SEM measurements of studied samples? Alternatively, are there antibunching measurements –assuming that they are quantum emitters?

From correlating PL and SEM images of the same sample region, we can conclude that most of the emissive diffraction-limited spots on the sample originate in fact from single objects. We note that our analysis does not allow us to exclude the presence of multiple emitters in some of the diffraction-limited spots. However, this should not be the case for the majority of the investigated objects.

Antibunching measurements are not available at this stage but will be part of future work. However, we would like to point out that there is no reason to expect antibunching in such large semiconductor crystals. They behave as pieces of bulk material where having two charge pairs does not lead to Auger recombination conditions (contrary to QD situation) simply because of the crystal volume is large and these charges have a lot of space and do not “feel” each other.

Implemented changes:

In the revised version of the SI we have carried out a detailed analysis of the SEM images (see **section 2**). In this section we discuss the overlapped PL and SEM images and we added a statement about the predominant occurrence of a single emitter per excitation spot.

- b.) How does the data in Figure 1 panel e change with total movie duration. If you analyze the data for 5 seconds total, 10 seconds total or 30 seconds total does the Arrhenius fit results change significantly? It would seem to the reviewer that there could be duration dependence of the overall ensemble intensity depending on the blinking statistics of individual emitters. For example, if you had exponential or power law distributed blinking, the longer one samples the greater the likelihood of encountering rare events.

As suggested by the reviewer, we plot the temperature dependent PL intensity for different time intervals of the intensity transients. The data is presented in **section 2 of the SM for review**. We do not observe significant variations of the slope of the Arrhenius plot and the fitted activation energy. It is certainly true that a longer integration time increases the probability of observing rare events. However, such events do not seem to have an influence on the time-averaged PL intensity.

c.) What are the circled points in Figure 1, panel e? This should be stated in the figure caption.

We thank the reviewer for this question. The circled points in Figure 1 indicate the temperatures, for which we show the intensity distributions in the adjacent panel (f).

Implemented changes:

We edited the figure caption to insert the missing information.

d.) Similarly, what is the excitation wavelength and excitation intensity for the data in Figure 1. Perhaps this should be stated in the figure caption or at least some mention of it made in the accompanying text.

The excitation wavelength was 458 nm and the ensemble-averaged excitation power density 0.3 W/cm² (see **SI, sections 2 and 3**).

Implemented changes:

We added a sentence to the figure caption in **Figure 1** to state the excitation intensity and wavelength.

e.) For the data in Figure 1 g-j, the reviewer is curious if there are accompanying spectra that show redshifts with decreasing temperature. Are the spectral energies of the emitter consistent on the cooling and heating paths? Part of the reason for the curiosity is that these spectral shifts would provide additional evidence for cooling and heating and if calibrated using say an ensemble measurement would provide local single emitter temperature estimates –as opposed to using the temperature controller suggested temperature. Would the Arrhenius behavior deviations then rectify themselves or become worse?

We have recorded temperature-dependent spectra for a few crystals of the original sample and provide this data in the **SI section 15**. The observed temperature dependence is similar to reported previously (see e.g. Milot et al., *Adv. Funct. Mater.* 2015, 25, 6218). However, even though there is a consensus in the literature about the temperature-dependent peak shift of the PL maximum, there are variations in the absolute values of emission energies and the temperature, at which the orthorhombic phase appears in luminescence spectra (compare e.g. with Kong et al., *Phys. Chem. Chem. Phys.* 2015, 17, 16405). Thus, we believe that calibrating the temperature via the emission maximum is not more accurate than using the temperature controller. However, we agree with the reviewer that showing the spectra is actually an important cross check for the correctness of the temperature.

Implemented changes: We added section 15 with the temperature dependent PL spectra to the SI.

f.) What is the excitation wavelength and excitation energy for the data in Figure 2? Again, this information should be stated nearby. The one thing the reviewer would like the authors to include in this discussion is the average number of excitations in the particle. Presumably, one wants to conduct blinking type measurements in a regime where Auger processes do not dominate. State also the assumed Auger coefficient. Additionally, temperature-dependent measurements should cause the absorption to shift. How much is the absorption coefficient changing between temperature and how does that impact the effective excitation densities in the nanocrystals?

The excitation wavelength and power density were the same as in **Figure 1**. To address the issue of the missing excitation power density, which was also raised by reviewer #1 (see reply to comment 1.1), we carried out a more detailed analysis of the crystal size to give a more precise estimate of the expected range of excitation power densities (**sections 2 and 3 of the SI**).

We thank the reviewer for the question about Auger recombination. To estimate the influence of Auger processes, we refer to literature reports, where the recombination coefficients of MAPbI₃ are stated (see e.g. Johnston and Herz, *Acc. Chem. Res.* 2016, 49, 146). With an Auger coefficient on the order of 10⁻²⁸ cm⁶s⁻¹, we assume that Auger processes do not play a role in the regime of excitation densities that we apply, but might

become relevant for excitation densities beyond 10^{17} cm^{-3} . Details of the discussion are given in **section 3 of the SI**.

For the absorption coefficient at a wavelength of 458 nm, an increase of 15 % is present in the data reported by Jiang et al., *Appl. Phys. Lett.* 2016, 108, 061905. However, applying Beer's law and considering a sample thickness of 70 nm, the number of absorbed photons changes only by about 6 % and we therefore neglect this effect.

Implemented changes:

In the caption of **Figure 1 (page 3)**, we indicate that the excitation conditions were the same for the measurements presented in the following figures. The estimations concerning the influence of Auger recombination are detailed in **section 3 of the SI**, where we also mention the effect of the temperature dependent absorption coefficient. Based on these estimates, we inserted a sentence on **page 10**, stating that Auger processes can be neglected in our experiments.

g.) How do the blinking trajectories change with excitation intensity below the onset of Auger? Does blinking frequency increase with increasing excitation intensity as might be expected? Conversely, does going to very low intensities increase the duration of on intervals at a given temperature?

A similar question about the dependence of the switching behavior on excitation power was also asked by reviewer #1 (see question 1.5). In short, we did not observe any clear dependence of the blinking dynamics or the ratio of times spent at bright and dim intensity levels on excitation intensity in our experiment (see also **section 7 of the SI**), even though this would be expected considering the hypothesis that blinking is related to light-activated ionic transport. However, as we discuss related to question 1.5, this may be due to the fact that the range of excitation densities applied in this study is actually higher than the range in which light activation of ionic transport has been observed. Note that all the excitation power dependent measurements were carried out below the onset of Auger processes, because we observe an increase of the PL quantum yield with increasing excitation power.

Implemented changes:

We included an updated version of the excitation power sweep with analysis of the blinking dynamics and discussion of photoactivation effects in the **SI (see section 7)**. In the main text, we briefly discuss the absence of photoactivation effects on **page 11**.

h.) How is the nanocrystal spectrum evolving during a given trajectory? Is the emission energy stable or is it changing with time indicating a spectral diffusion process linked to the intermittency?

We did not perform an extensive study of the evolution of the spectra, but studied some individual crystals at low temperatures, because we were especially interested in the question if both the tetragonal and orthorhombic phase reveal intensity fluctuations. It turned out that both the low-energetic signature of tetragonal phase and the higher-energetic peak related to the orthorhombic phase reveal blinking. Some exemplary data is shown in the SM for review (**section 3**). By plotting spectra associated with different intensity ranges of a transient, we find that in some cases the intensity change is indeed related to a shift of the emission maximum. A red-shift of the spectrum in the brighter state compared to the spectrum in the darker state could be due to the fact that photoexcitations have more time to relax spectrally and emit from lower-energetic states when the quencher is passive. On the other hand, energetically different emission maxima could also indicate that the emission in the bright and dim state comes from spatially separated regions with different local properties. Note that a remarkable shift is only observed in one of four cases. We feel that spectral changes are beyond the scope of the current manuscript. A more detailed study of the evolution of the emission spectra is planned for future work.

i.) Do the individual nanocrystals eventually photobleach?

The effect of photobleaching in vacuum and especially at low temperature is very limited at the given excitation conditions. In the **SI (section 1)**, we now plot the temperature dependent PL intensity separately for the heating and cooling cycle. The temperature-dependent experiment was carried out in a way that the sample was first cooled down from room temperature to 80 K and then heated up again. Thus, any influence of

photobleaching should, at the best, become apparent in a comparison between the PL intensities at 295 K, measured for the heating and for the cooling cycle, respectively.

As a result, we find that the intensity of the heating cycle resembles that of the cooling cycle in most cases. Discrepancies between the two cycles do not indicate a systematic photobleaching and can rather be explained by the presence of blinking. However, we note that at room temperature the time-averaged PL intensity of most of the crystals in the heating cycle is below the corresponding value in the cooling cycle. This could be an indicator that photobleaching under room temperature should be taken into account, even though the crystals reveal a high photostability at lower temperatures.

Implemented changes:

In **Figure S1 in the SI**, we now show the PL intensities of the heating and the cooling cycle separately, instead of averaging between the two. From this, it becomes apparent that photobleaching after long-term exposure to the laser can be neglected.

j.) What is the estimated emission quantum yield of an individual nanocrystal during a given on period at different temperatures? This can be estimated knowing the temperature dependent absorption coefficient of the particle and the estimated collection efficiency of the instrument. Is unity quantum yield ever achieved? A corollary question would be what the lifetime traces looked like during on and off intervals. Presumably during stable on portions of a trajectory one would see an exponential decay which together with the quantum yield estimate would enable one to suggest what radiative and non-radiative rate constants were involved? Is this data available as it would help.

In response to comment 2.7.a) we have estimated the temperature dependent luminescence yield for two individual crystals (see **SI section 4**). For these particular examples it turns out that even at 77 K the PL quantum yield is only around 20 %. To determine the luminescence yield it is important to know the crystal size, local excitation power and the amount of emitted light at the same time. Moreover, for estimating the absolute decay rates we have to know the decay times during ON and OFF periods. Unfortunately, for none of the crystals both the luminescence yield and the time-tagged decay times are available at the same time.

k.) For temperature dependent measurements, what is the exciton versus free carrier fraction in the nanocrystals, assuming that these crystals are outside their confinement regime, that dielectric contrast does not enhance bulk binding energies and also employing the Saha equation. Next, there are presumably a larger fraction of excitons present at low temperature while relatively few at room temperature. Are the authors suggesting that the quencher sites do not distinguish between free carriers and excitons? If not, are the authors suggesting subcategories of quenchers that should be introduced into subsequent kinetic modeling?

We are grateful for this question which forced us to look at this in more details. As suggested by the reviewer, we used the Saha-Langmuir equation to estimate the temperature-dependent fraction of the excitonic population in the crystals. Details of the analysis are available in **section 13 in the SI**. It turns out that at room temperature, the fraction of excitons is relatively small, most likely below 10 %. At temperatures as low as 80 K, however, an excitonic fraction on the order of 50 % is not unrealistic (the uncertainties here are due to lack of consensus on the exciton binding energy in MAPbI₃) It is therefore important to discuss whether the growing excitonic population at low temperatures could be connected to some of our experimental results.

In short, we think that growing exciton population alone cannot explain the observed decreasing of the quenching efficiency. Moreover, even if excitons cannot be quenched by the same non-radiative center as a pair of free charges (which is unlikely to our opinion), the quenching can happen during the time the exciton is dissociated due to the presence of dynamic equilibrium between excitons and free charges.

Implemented changes

The full analysis based on the Saha-Langmuir equation including a detailed discussion of the role of excitons has been added to **section 13 of the SI**. In addition, we believe that it is important to include our estimations of the exciton versus free charge population and parts of the above made discussion in the manuscript, because the current version does not specify which photoexcited species is actually present. We inserted a corresponding passage in the discussion on **page 11-12**.

l.) The labeling in Figure 4 must be improved. It is not clear which trace is Figure 4b. It was also unclear to the reviewer how the switching rate was estimated for this Figure. Perhaps some mention of this would improve things.

We thank the reviewer for the close reading. In fact, Figure 4(b) does not exist and the reference to panel (b) unintentionally remained from a former version of the manuscript. The switching rate was obtained from counting blink events above a certain threshold from the model trace and dividing this number by the measurement time.

Implemented changes:

We corrected two passages in the text, where Figure 4(b) was mentioned on **page 6** and **page 11**. In addition, we added a sentence to the caption of **Figure 4**, in order to give a more comprehensive description how the switching rates were determined.

m.) In equation 2, it was not clear why physically each quencher needed to have the same decay rate constant.

We agree with the reviewer, that there is no any necessity of this, except for the sake of simplicity. Assuming the same rate constant for all fluctuating quenchers allows us to simplify equation 2 and define the temperature dependent parameter $\kappa(T)$, which describes the ratio between the fluctuating and the static decay rates. This parameter simplifies our simulation and we find that the model, in its current version, reproduces the essential features of the experimental observations. However, as the reviewer points out, the text suggests that there is a physical reason why all the quenchers reveal the same quenching rate. This is probably not true, since variations in the local environment of a quencher and inhomogeneities in the material rather point towards a distribution of the quenching rate. Since we do not know how this distribution looks like, we decided to use the simplest approach that reproduces our observations.

Implemented changes:

In order to clarify that the assumption of a single quenching rate is a simplification made in the model, we modified the text on **page 7**.

n.) The authors should probably define E_a , E_x and E_p close to Figure 3 where these parameters are introduced.

The parameters E_a , E_p and E_x are introduced after equation 3 on **page 7**. However, to make it more visible we inserted a bit of text on **page 7** to clarify the meaning of these parameters. Additionally, the meaning of these parameters is explained in the figure caption and they are shown in the new **Figure 5(a)**.

o.) The reviewer thought that the ion migration hypothesis was interesting but did not understand it. Perhaps this should be explained in some more detail so as to establish within the context of the hypothesis what an active defect configuration looks like versus a passive one. Along the same lines and as suggested by the authors -why would ion migration necessarily be reversible in the sense of cooling and heating trajectories having the same quencher configuration.

We thank the reviewer for this important comment and agree that the original version of the manuscript did not provide a clear picture about how ion migration is involved in the switching of the defects between an active and a passive configuration.

Implemented changes:

In order to make the suggested mechanism clearer, we re-arranged the corresponding two paragraphs on **page 11**. Moreover, instead of showing different configurations of mid-gap energy levels in the original Figure 6(c) and (d), we added a scheme of our proposed mechanism for the switching between a passive and active defect configuration, here exemplified for a donor-acceptor complex, which is now **Figure 7(a)**. Following the suggested picture, reversible switching properties and PL intensity would arise due to the fact that the efficient quencher is actually fixed in space and temperature dependent migration of ions governs its switching properties (see Merdasa et al., *ACS Nano* 2017, 11, 5391).

p.) Next, carrier mobilities are quite large in hybrid perovskites. It would seem that the physical act of a carrier diffusing to a local quencher and extinguishing the excitation would be very efficient. Perhaps too efficient. The

timescales of carrier diffusion to a site within a 50 nm distance must be extremely short. Why would sustained on times be even possible with any regularity given multiple quenchers per nanocrystal?

Here, it is important to note that the quenchers in many crystals we are observing spend most of their time in passive configuration. Thus, there are time intervals, during which none of the quenchers is active, such that the observation of extended on periods is generally possible. In addition, even if a quencher is active, there is some residual PL intensity. It is true that diffusion towards the quenchers is very efficient, but our correlation of the intensity of a blinking transient with the corresponding PL lifetimes has revealed that in such cases where diffusion does not limit the quenching efficiency, quenching is limited by the capture cross section (see **section 14 of the SI**). Thus, even if one quencher is active during the whole time of the measurement, a second quencher could still modulate the residual PL intensity in a way that pronounced on and off switching would be observed.

However, the situation becomes different if more than one or two of the fluctuating quenchers are active or if they switch as fast that their contribution can be treated as an effective quenching rate. Such behavior affects the PL intensity in particular at room temperature, where the quenching efficiency of the fluctuating quenchers is generally higher than at low temperatures. Actually, such a situation might apply to most of the crystals at room temperature. For our analysis, however, we only chose crystals with intensities that reveal a sufficiently high signal-to-noise-ratio (SNR) for all temperatures of the cooling and heating cycle. Our method is thus more selective to the crystals with a smaller amount of active quenchers at room temperature, whereas another subset of the group simply does not appear at higher temperatures.

Implemented changes:

In order to illustrate this behavior, we plot the temperature-dependent number of crystals with intensities above a certain signal-to-noise ratio (see **SI, section 16**). This number increases roughly by a factor of 3, when the crystals are cooled from room temperature to a temperature of 80 K. Thus, we believe that a certain subset of the crystals reveals in fact a very low PL quantum yield at elevated temperatures, as expected by the reviewer. In the manuscript, we refer to this new section in the SI on **page 3**, in connection with mentioning the high variation of PL intensities observed at room temperature.

q.) It was not clear how the early ensemble 63 meV activation energy linked up to the 0.2-0.8 eV single quencher barriers suggested from subsequent temperature-dependent measurements.

We thank the reviewer for pointing this out, indeed it was not clearly explained. The two activation barriers mentioned by the reviewer actually describe two different processes. The value of 63 meV was obtained from fitting the temperature dependent PL intensity and is an ensemble-averaged measure for the activation of non-radiative processes in the material. The energy range from 0.2 to 0.8 eV on the other hand is an estimate for the activation energy of the switching of a non-radiative center between passive and active state. It is thus related to the defect chemistry and does not reveal information about the quenching efficiency.

Implemented changes:

We understand that the mentioning of two activation energies could actually be misleading. Thus, in connection with the introduction of the activation barrier for switching of quenchers on **page 8**, we inserted a sentence to emphasize the distinction of these processes.

r.) The authors say that they adjusted the parameters of the simulation (such as the number of quenching sites to ~1-3 per nanocrystal) to improve the agreement with the experiment. What features of the time trajectories were improved from this? It appears the main feature is the intensity histogram. Did time-correlation features such as on and off-time distributions or power spectral density improve?

With our simulations, we essentially tried to mimic the temperature dependencies of the PL intensity and the switching rate. We did not look into on and off time distributions or the power spectral density function. The main reason for that is that on-off-analysis cannot be really applied to such complex blinking transients, because it is impossible to define a meaningful threshold for the discrimination between bright and dark states. However, for future work it would be desirable to further work on improving the model and define more criteria to match the experimental with the simulated data. So far, the intention of our simulations is to give a

'proof of concept', to demonstrate that our model indeed describes the relevant switching and quenching processes.

Reviewer #3

3.1. The title is "Microscopic insight into non-radiative decay in perovskite semiconductors from temperature-dependent luminescence blinking". However, in this paper, the system under investigation is based on perovskite NPs. It is well known that during the formation of NPs, there exists a large amount defects and surfaces, which make them entirely different from the property of periodical structure crystals. It might be difficult to justify that the same mechanism is still valid in perovskite single crystal or even perovskite thin film.

We thank the reviewer for this comment and we understand the reviewer's concerns about the comparison of properties of small crystals and films. Indeed, it is important to explain better the relevance of our work to a broader audience. First of all, we note that we are dealing with rather large particles (not what is usually termed "nano crystals" – 10 nm quantum dots). Our sample is comprised of crystals of about 70 nm in diameter. The blinking behavior as such has been reported in MAPbI₃ for objects of sizes up to several micrometers, which means that it is definitely not an exclusive behavior of small particles, but rather a very general phenomenon. In the light of growing number of studies on blinking in diverse perovskite systems and very diverse scale (from small nanocrystals to microcrystals and parts of intact films) and our own experience, we strongly believe that blinking in these materials is actually a universal phenomenon and must be related to the intrinsic features, for example to ion migration as we propose here. Therefore, although we agree with the reviewer that there must be quantitative differences due to e.g. larger surface effect between our crystals and intact film, we think that the difference is quantitative, but not qualitative.

In order to corroborate the versatile appearance of blinking and a broader validity of the observations made in our samples, we will therefore in the following text refer to publications of other groups together with our own studies.

A general measure for the presence of non-radiative decay channels in a semiconductor is the enhancement of the PL intensity upon cooling. For the nanocrystals studied here, we found an enhancement by about two orders of magnitude between 295 K and 80 K, in agreement with previous reports on thin films (see e.g. Savenije et al., *J. Phys. Chem. Lett.* 2014, 5, 2189; Fang et al., *Light Sci. Appl.* 2016, 5 or Stranks et al., *Phys. Rev. Appl.* 2014, 2,1) or even some work on single crystals (see Fang et al., *Adv. Funct. Mater.* 2015, 25, 2378). From this and from analyzing the PL decay, we infer, that the amount of non-radiative channels in our nanocrystals is similar to perovskite films. This is also in line with our estimated density of the fluctuating quenchers of about 10¹⁶ cm⁻³, which matches the numbers reported for thin films (see discussion on page 8 of the manuscript). In addition, we note that perovskite films are typically comprised of individual sub-micrometer-sized (typically 100 – 500 nm) grains revealing defect sites with a varying influence on non-radiative decay (see e.g. deQuilettes et al., *Science* 2015, 348, 6235). Therefore, we do not expect them to behave substantially different from the nanocrystals studied here, given that no other optimization procedure such as surface passivation has been carried out.

Single crystals on the other hand are expected to comprise much lower defect densities due to the absence of internal grains and the usage of capping agents (see e.g. Yuan et al., *ACS Omega* 2016, 1, 148). They are therefore well-suited to study the intrinsic material properties, but not necessarily representative for the actual processes in perovskite thin films or devices. However, even in large single crystals (for example 9 micrometers long, 500 nm in thickness) PL blinking was clearly observed (see Merdasa et al., *Nano Lett.* 2015, 15, 1603).

Implemented changes:

In response to the comments 3.1-3.3, we have added a passage at the end of the discussion section (**page 13**), where we stress that blinking has been observed in a variety of perovskite samples and preparation routes. Therefore, we consider it as a relevant decay mechanism which has to be suppressed for further improvement of the material quality.

3.2. In the sample preparation part (Page 11), the authors claim that the samples were fabricated in ambient atmosphere. It is widely accepted that the humidity and oxygen molecules have significant impact on the

performance of perovskite device, especially in the perovskite NPs, which possess large surface area to volume ratio. Is there any difference between the device fabricated in N₂ filled glovebox and ambient condition?

We agree with the reviewer that the processing conditions and the environment are very important for the electronic properties. In our case, it is only the preparation which was done in ambient conditions, while all experiments were done in high vacuum. Considering the ease of diffusion of oxygen and other gases through MAPI films, we think that our experiments are done in conditions (high vacuum) where the environmental effects are absolutely minimal. As for the preparation itself, we have not tested this particular preparation technique (low concentration of the DFM:DMSO precursor) for differences in blinking behavior. Again, we expect that there will be a difference, but we expect this difference to be only quantitative (higher or smaller concentration of the switching non-radiative centers), most probably different PL lifetime which we think is a consequence of other non-radiative recombination channels (the parameter Φ_0 in equation 2). Therefore, such comparison is definitely an interesting study by itself, but we think it goes beyond the current work.

Regarding the differences in the processing conditions, it is important to mention that blinking has also been observed in samples that were prepared in a glove box, i.e. in the absence of oxygen and humidity (see e.g. Yuan et al., *Adv. Mater.* 2018, 1705494 or Li et al., *Molecules* 2016, 21,1). Yuan et al. specifically investigated different preparation conditions and reported that they did not find any influence of the atmosphere on the formation of the quenching sites that cause PL blinking. Thus, we have no doubts that PL blinking would also be observed, if the samples were prepared in a glove box.

Despite this apparent insensitivity of the presence of PL blinking as such to the processing conditions, it should not be denied that the environment has been demonstrated to have a certain influence on the emergence of blinking. Yuan et al. (*ACS Omega* 2016, 1, 148) have reported different appearances of blinking in a cluster of nanocrystals in different environments (ambient air, nitrogen and vacuum) and thus attributed the observed blinking to surface traps. Similar to their work, we observed blinking in our samples in ambient air and vacuum. Here, we do not provide a definite answer to the nature of the observed traps, but irrespective of their occurrence either in the bulk or on the surface we argue that it is highly likely that the same defect species is also relevant for perovskite films and devices.

Implemented changes:

In response to the comments 3.1-3.3, we have added a passage at the end of the discussion section (**page 13**), where we stress that blinking has been observed in a variety of perovskite samples and preparation routes. Therefore, we consider it as a relevant decay mechanism which has to be considered for further improvement of the material quality.

3.3. I wonder whether the authors have observed the similar blinking behavior in compact perovskite thin film.

This is a very relevant question indeed which we admittedly did not address in the original submission. Blinking has also been observed in MAPbI₃ films that were prepared in a glove box (see Li et al., *Molecules* 2016, 21, 1). Blinking in a MAPbBr₃ film has been reported by Wen et al., *Nano Lett.* 2015, 15, 4644. We ourselves observed PL blinking in MAPbI₃ films which confirms the study by Li et al., *Molecules* 2016, 21, 1. The main reason why PL blinking is usually not studied in films is that because it much less pronounced due to obvious ensemble averaging effect when different crystallites of the film fluctuate independently (see Li et al).

Implemented changes:

We give the reference to the observation of blinking in perovskite films (Li et al.) in the discussion on **page 13**.

3.4. According to Galland et al. [*Nature* 479, 203(2011)], it is known that there are two kinds of PL blinking, (1) conventional one due to charging and discharging, (2) blinking due to charge fluctuation in the sites. By investigating the PL intensity against the lifetime, it is possible to distinguish these two types of blinking mechanism. I wonder whether it is possible to present this result.

Yes, we agree with the reviewer that PL lifetime can either change in correlation with changing the PL intensity or stay the same in spite of PL quenching. In order to answer the reviewer's question and some previous remarks (1.3, 2.5 and 2.7(j)), we recorded time-tagged blinking transients with an APD detector, from which we can extract the PL decay dynamics and relate it to different emission intensities. Details can be found in **section**

14 of the SI. In their study on quantum dots, Galland et al. have found different blinking behavior and related this to two different mechanisms: In one case, the change in PL intensity was related to a change in the PL lifetime. This type of blinking was attributed to the commonly assumed charging and discharging of the nanocrystal shell. In the second case, a change of the PL intensity did not result in a change of the PL lifetime, which was explained by a very fast hot electron capture process of a metastable non-radiative center.

The data we collected did not allow us to produce correlation plots of a quality (especially at low temperature) comparable to those presented by Galland et al. However, importantly, our results do not indicate the presence of two distinct quenching mechanisms. In all investigated cases, we observe a correlation between the PL lifetimes and the PL intensity. This correlation is, in most cases, less strong than what would be expected for the case of a purely dynamic quenching mechanism. We explain this observation by a partially diffusion-limited quenching process. In contrast to the report of Galland et al., we believe that for the case of our crystals, the observations can be consistently explained by a single quenching mechanism through the metastable quenchers, which is partially diffusion-limited. Note that in contrast to quantum dot systems, luminescence quenching due to an Auger process appears unlikely because the excitation densities are far below the Auger regime, see our response to comment 2.7(f) and **SI, section 3**.

Implemented changes:

The data on PL lifetime vs PL intensity is presented in **SI section 14**, also see discussion on **page 12** in the main text. The paper by Galland et al and other relevant papers to this context are cited.

3.5. The excitation intensity in the experiment is 0.3 ± 0.2 W/cm². This is a very high intensity which might lead to the increased temperature of the sample surface. Please comment on the possible influence.

We agree with the reviewer that our excitation intensity is not very low, however it is about solar flux which is 0.1 W/cm². This excitation power is well inside the regime of intensities, which are typically used for optical experiments also with perovskite films. We also note that the size of the excitation spot (about 50 μm) is very small compared to the substrate, so we expect efficient dissipation of heat into the substrate. A remarkable influence of the laser power on heating of the substrate should occur for laser powers, which are about three orders of magnitude higher than the intensities applied here (see e.g. Lo and Compaan, *J. Appl. Phys.* 1980, 3, 1565).

Moreover, in response to comment 2.7(e), we present temperature dependent PL spectra in **section 15 of the SI**. The temperature-dependent energetic shift of the emission maximum and the peak shift related to the phase transition are consistent with previous work, thus, we believe that the temperature measured by the instrument is actually close to the temperature of the sample under laser excitation in our experiments.

3.6. The authors attribute the blinking behavior to the ions as the quencher. It is important to consider the influence of photogenerated charges under light illumination, as well as the interaction between the photogenerated charge carrier (electrons and holes) and ions (defects).

We thank the reviewer for this comment. We have investigated the effect of excitation density on the quenching efficiency (blinking amplitudes) and the switching rates of the quenchers (see **section 7 in SI**). Within the range of applied excitation densities, we do not observe any substantial influence of the excitation power density on the switching behavior of the studied quenchers. Light activation of ion migration is a well-known phenomenon in perovskite semiconductors, however, we believe that the effect is rather apparent at excitation densities below the range applied in this study (see Zhao et al., *Light: Science & Appl.* 2016, 6, e16243). Another effect could be saturation of the quenchers, when too many charges are present. This has indeed been observed by Tian et al. (*Nano Lett.*, 2015, 15, 1603), but not in our case, which we attribute to the smaller crystal size compared to previous work. For a more detailed discussion about the influence of the excitation density we refer to our response to comment 1.5.

Implemented changes:

The influence of different excitation intensities and, with this, different carrier densities is mentioned in the revised version on **page 11**. Moreover, we included the updated version of the excitation power sweep in the **SI** of the manuscript (see **section 7**). There, we also added our investigation of the influence of photoactivation and a brief discussion of the results.

3.7. Figure 1(g) to (h) depict the temperature dependent PL. However, in the method part, the authors did not provide any experiment details, such as the heating speed, cooling speed, in dark or continuing illumination, etc. In addition, Figure 1(g) to (h) shows negligible hysteresis during the heating and cooling process. However, in Fig S7, cr 193, cr 190 and cr 175 exhibit significant hysteresis behavior. Please give the comment on the behavior.

We thank the reviewer for careful reading of our paper. Indeed, we did not provide enough experimental details in a few places. Most importantly, we did not illuminate the sample during the temperature change and stabilization. After reaching a certain temperature, we waited for five minutes before starting the measurement to let the system stabilize.

We believe that the hysteresis behavior in the switching rates that was mentioned by the reviewer comes from an effect that we refer to as the ‘freezing’ of fluctuating quenchers in the regime of low temperatures. We thank the reviewer for paying attention to these few examples. In the revised version we use these crystals as examples when we discuss the idea of “freezing” (see **page 13**).

A hysteresis in the temperature dependence of crystals 175, 190 and 193 can also be seen in their temperature-dependent PL intensity plotted in **section 1 of SI**. Based on the proposed model for temperature-dependent blinking, we assume that switching in the low-temperature regime becomes very slow. For some quenchers, it becomes thus very unlikely that they produce blink events during the duration of the measurement. However, to a certain extent, such events still do occur and can lead to different behavior in the heating and the cooling cycle, e.g. if a quencher is active during one cycle and passive during the other.

Implemented changes:

In response to the remark on the experimental conditions during the temperature sweep, we added some more specific information about the temperature sweep to the methods section on **page 16**.

In the manuscript we refer to the phenomenon of ‘freezing’ in the discussion of the model on **page 13**. We edited this section, in order to include the experimental observation of hysteresis behavior, which, as we believe, is a consequence of freezing of the quenchers in different states. In order to demonstrate the hysteresis behavior also for the PL intensity, we now show the temperature-dependent PL intensity separately for the heating and cooling cycle in the supporting information of the manuscript (**Figure S1**).

3.8. On page 8, the authors indicate that the activation energy is between 0.2 and 0.8 eV. Additionally, the authors mention that 300 K correspond to 0.025 eV. Hence, it is nearly impossible to overcome the barrier only by the room temperature. In addition, the authors said they would provide detailed discussion on section 10, but I did not find the discussion in SI. Please provide detailed information and discussion.

We thank the reviewer for pointing this out and we realized that several important aspects were not explained clear enough in the original text. It is indeed not intuitive, why blinking in the investigated temperature range should emerge, when the energetic barriers for switching E_0 are on the order of several hundreds of meV which is much larger than kT . Here it is important to bear in mind that the probability for switching per unit time does not only depend on the energetic barrier, but also on the attempt rate for switching. If the barrier height is about kT , then even one attempt to jump over can be successful. If the barrier is higher, however, the system needs to try many more times making the transition slow. Here the question is how slow and in comparison to what. Since we expect this attempt rate to be quite high (typical vibration frequency in solids, 10^{12} s^{-1}), while switching occurs with frequencies of about 1 s^{-1} or even less, after 10^{13} attempts at room temperature a barrier of even 1 eV can be overcome once (see eq. 3 of the manuscript).

Later in the model section of the original submission, we briefly discussed the energetic difference ΔE between the two switching configurations and we stated that this difference cannot be too large when blinking is observed, because otherwise the system would reside in one of the states for most of the time. Related to that, we referred to the supporting information for a more detailed discussion. However, there was a mistake in the section number to which we were referring; it should be section 9 in the original version of SI and not section 10. We apologize for this confusion. Based on the reviewer’s comment we realized that the overall explanation still needed a lot of improvement, thus we performed some more extended revisions of this section and some parts of the original SI became part of the revised manuscript.

Implemented changes:

In order to explain the seemingly contradictory statement that energetic barriers much higher than kT lead to blinking at room temperature, we modified the description of the model on **page 8**, where we now explain in greater detail how energetic barriers on the order of several 100 meV lead to switching rates that can be observed experimentally by pointing out the role of the attempt rate. We also added an example calculation of the switching rate employing equation 3 to the corresponding passage.

Moreover, we decided to show the scheme of the energetics for switching between active and passive state and the outcome of the Monte Carlo simulations in two separate figures (now **Figure 5** for the model and **Figure 6** for the simulations). In addition to the energy schemes, Figure 5 also contains two plots based on which we discuss the model parameters E_0 and ΔE in greater detail, which has previously been done in the SI.

These changes also imply that we removed certain sections from the originally submitted supplemental material, since these issues are now treated in the main text: Figure S9 and related discussion, as well as SI section 9.

3.9. From Page 17 to Page 18, I cannot find Figure 7 to Figure 12 in the main text.

We apologize for these misprints. The figure captions in the end of the document are just a repetition of the captions of the figures in the manuscript and the higher numbers resulted from automated numbering which we did not check.

3.10. In Figure 6(b), E1 and E2 should be $\Delta E1$ and $\Delta E2$.

We changed the mistake in Figure 6(b) (now Figure 6(c)).

3.11. The authors should be careful with the Reference part. Ref (8), (25), (42), (54), (58), (59), (61) are not properly cited.

We thank the reviewer for the careful reading and corrected the corresponding references.

Changes beside the ones described above:

Based on the reviewer's comments we performed an extensive revision of the supporting information and references to the supplemental material in the main text have been updated.

Abstract: The abbreviation "OMHP" has been removed, because it was not used in the abstract.

Page 2: Two recent and relevant papers on luminescence Blinking in MAPbBr₃ crystals are now cited in the section of the introduction, where we refer to related work on blinking in perovskites (Pathoor et al., *Angew. Chemie - Int. Ed.* 2018, 57, 11603–11607, Yuan et al., *Adv. Mater.* 2018, 30, 1705494).

Page 3:

- The reference to Figure 1(e) in the text should actually refer to Figure 1(f), and the reference to Figure 1(f) to Figure 1(e). The issue has been corrected.
- There was a typo in the caption of Figure 1. The last sentence should refer to panel (k) and not to panel (f).

Page 4: Introduction of the abbreviation "SI" for supporting information.

Page 7: There was a mistake in equation (3). The rate $k_{p \rightarrow a}$ is related to the energy E_{px} and $k_{a \rightarrow p}$ depends on E_{ax} . In the original manuscript, we confused the indices.

Page 9: Sentence reformulated:

"...The decrease of $\kappa(T)$ upon cooling can come from a higher radiative rate ~~at low temperature~~, a ~~smaller~~ lower non-radiative rate per quencher or a combination of both factors. ..."

Page 13: In the last paragraph, we refer to the strong variations observed for the PL enhancement. Here, the correct reference is to figure 1(g-j), not Figure 1(e).

Figures 1 and 3 and the caption of Figure 3 have been updated according to the data presentation guidelines in the editorial policy checklist. Moreover, we added a data availability statement.

Reviewers' comments:

Reviewer #1 (Remarks to the Author):

In this revised manuscript, authors have well addressed reviewers' comments. Accordingly, they carried out further experiments, added deep discussions on various aspects of the microscopic nonradiative mechanism and presented detailed experimental conditions. Overall, they significantly improve the manuscript. This manuscript has provided new insight on the microscopic mechanism on carrier recombination dynamics, although there is some inconsistency due to inhomogeneity and randomness in individual nanoparticles. This manuscript can be a very useful reference for further comprehensive understanding the mechanism of carrier recombination and interaction between photogenerated carriers and mobile ions in perovskites. I suggest authors consider a further question: the species of the quencher? based on the current evidence, together with previous reports on the most likely mobile ions (Eames et al. Nature Communications 6, 7497, 2015, and many other literatures) and selective charged mobile ion induced quenching (Chen et al. ACS Appl. Mater. Interfaces 8, 5351, 2016; Deng et al. J. Mater. Chem. C, 4, 9060, 2016). I think this manuscript can be considered for publishing in Nature Communications.

Reviewer #2 (Remarks to the Author):

This manuscript came a very long way, and at this stage I think is ready for publication. Given the extensive revisions and additions required after the comments of the referees, it is clear that the original manuscript was nowhere near the level of finalization required by a leading publication, such as Nature Comm. However, the results reported in this paper remain very interesting and very important. I recommend publication.

Reviewer #3 (Remarks to the Author):

In this revised manuscript, authors have significantly enhanced the interpretation. I think authors have reasonably addressed referees' concerns and questions and this manuscript has been improved. However, authors still need to address some more concerns before it can be published:

1. On Figure S2, by using the temperature dependent PL intensity, authors obtain the activation energy of 63 meV. Generally, temperature dependent PL rising rate is used to obtain the activation energy. [Energy Environ. Sci., 2016, 9, 3180--3187]. Is it possible to provide more comments on the comparison?

In addition, I am confused with the interpretation of the activation energy. The author mention that "In terms of our model, 63 meV does not correspond to any energy difference between real states, it results from the convolution of switching of the fluctuating quenchers, their strength described by and persistent quenching characterized by Φ ." Please provide more comment on this statement.

2. The author proposed that ion migration is the main reason for the PL blinking behavior. In addition, authors mention that the ion migration is photoactivated. It is necessary to explain the detail process, that is, how the photons absorbed by perovskite materials drive the ionic movement, and which is the main driven force. It is noted that, although the illuminated visible light equips with the similar or even higher energy (several eV) than the activation energy of ion migration, these photons cannot directly move these ions.

3. The author cited "Molecules 21, 1–12 (2016).", it should be: " Molecules, 21, 1081 (2018)"

4. Recently, it is proposed that the oxidation state of halide vacancy may play an important role in nonradiative charge recombination. [J. Am. Chem. Soc. 2018, 140, 15753–15763]. It may be helpful to understand the possible PL quenchers in perovskite.

We thank the reviewers for their time spent with this manuscript and for their constructive comments and suggestions. Our detailed response is given below. Important changes in the manuscript are highlighted by blue font.

Reviewers' comments:

Reviewer #1 (Remarks to the Author):

In this revised manuscript, authors have well addressed reviewers' comments. Accordingly, they carried out further experiments, added deep discussions on various aspects of the microscopic nonradiative mechanism and presented detailed experimental conditions. Overall, they significantly improve the manuscript. This manuscript has provided new insight on the microscopic mechanism on carrier recombination dynamics, although there is some inconsistency due to inhomogeneity and randomness in individual nanoparticles. This manuscript can be a very useful reference for further comprehensive understanding the mechanism of carrier recombination and interaction between photogenerated carriers and mobile ions in perovskites. I suggest authors consider a further question: the species of the quencher? based on the current evidence, together with previous reports on the most likely mobile ions (Eames et al. Nature Communications 6, 7497, 2015, and many other literatures) and selective charged mobile ion induced quenching (Chen et al. ACS Appl. Mater. Interfaces 8, 5351, 2016; Deng et al. J. Mater. Chem. C, 4, 9060, 2016). I think this manuscript can be considered for publishing in Nature Communications.

Reply 1. We thank the reviewer for the very positive view on the revised manuscript and also for the fruitful suggestion. Indeed, so far we have been rather reluctant to discuss candidates for the quenchers, since our own study is not able to draw direct conclusions about their chemical origin. However, we agree that giving a more detailed picture considering other work is possible. In the previous version of the manuscript we have already indicated that ion migration is probably involved in the switching mechanism. In the revised version we explicitly say that iodide and hydrogen species are the most mobile and need to be considered as first candidates for triggering the switching of the non-radiative channels. The publications suggested by the reviewer are cited now in this context (Refs 42, 56, 57).

We favor the idea of complexes of simple defects as strong non-radiative channels because their dissociation can be easily connected to the temperature dependent ion diffusion. Moreover, each defect individually can be absolutely benign which fits to the defect tolerance concept.

However, reviewer #3 pointed out to some recent computational studies suggesting that deep energy levels inside the band gap could also be provided by a single species, e.g. interstitial iodine (Mosconi et al., Energy Environ. Sci. 2016, 9, 3180) or anionic iodine vacancies (Li et al., J. Am. Chem. Soc. 2018, 140, 15753–15763), we mention this in the revised discussion. However, one needs to come up with a mechanism of their switching. So far this problem has never been addressed theoretically neither for point defects nor for complexes and we hope that our study will give some inspiration for this.

Finally, inspired by this comment of the reviewer we polished the whole discussion section on pages 10-12. The important changes are highlighted by blue font.

Reviewer #2 (Remarks to the Author):

This manuscript came a very long way, and at this stage I think is ready for publication. Given the extensive revisions and additions required after the comments of the referees, it is clear that the original manuscript was nowhere near the level of finalization required by a leading publication, such as Nature Comm. However, the results reported in this paper remain very interesting and very important. I recommend publication.

Reply 2. We are grateful to the reviewer for the very positive evaluation of our work.

Reviewer #3 (Remarks to the Author):

In this revised manuscript, authors have significantly enhanced the interpretation. I think authors have reasonably addressed referees' concerns and questions and this manuscript has been improved. However, authors still need to address some more concerns before it can be published:

1. On Figure S2, by using the temperature dependent PL intensity, authors obtain the activation energy of 63 meV. Generally, temperature dependent PL rising rate is used to obtain the activation energy.[Energy Environ. Sci., 2016, 9, 3180--3187]. Is it possible to provide more comments on the comparison?

Reply 3A

We thank the reviewer for this question and for pointing to the very relevant literature. In the paper mentioned by the reviewer it is the temperature dependence of the rate of the PL enhancement that is connected to ion migration. The underlying hypothesis in this work is that defect curing by light is related to ion migration. We would like to stress, that this experiment is very different from the PL enhancement upon cooling that we discussed in our manuscript and cannot be compared with it. We absolutely agree that the activation energy obtained in [Energy Environ. Sci., 2016, 9, 3180--3187] can indeed be assigned to ion migration, however, the 63 meV energy obtained from formal fitting of the PL intensity (T) cannot. Accordingly, in the revised version we cite this work (Ref. 55) on page 11, bottom, while referring to the range of activation barriers that has been reported for the migration of different ionic species.

We stress, that the intention of the fit in Figure 1 was not to make any statement about ionic transport. The fit simply allowed us to compare the slope of PL(T) to other studies, where similar values have been reported but for bulk films rather than individual nanocrystals. Moreover, we used the fit to demonstrate that the parameter ε depends strongly on the incorporated defects, which can be seen from the fits of individual crystals in Figure 1(g)-(k). Since the fitting procedure is simple and therefore often used in the literature, we found it important to point out this aspect to the community, to avoid misinterpretations of the fitting in future work.

In addition, I am confused with the interpretation of the activation energy. The author mention that "In terms of our model, 63 meV does not correspond to any energy difference between real states, it results from the convolution of switching of the fluctuating quenchers, their strength described by and persistent quenching characterized by Φ ." Please provide more comment on this statement.

Reply 3B.

We thank the reviewer for pointing out again on this confusion. We realized that in the figure 1 and in SI in Figure S3 we used a wrong notation (E_a) for the phenomenological fitting parameter which

should be called ε instead in order not to mix it with the activation energy of the quencher switching. This could be one reason for the confusion. In the revised version the Arrhenius fitting parameter is called ε everywhere.

The purpose of the section was to avoid confusion between the Arrhenius fitting parameter $\varepsilon = 0.063$ eV and the range of energies of $E_0 = 0.2 - 0.8$ eV, which is the estimated energy range for the activation of blinking, based on our model on p. 7. The fitting parameter ε describes thermally activated non-radiative decay without any connection to microscopic processes. In the framework of our model based on the experimental results, this energy indeed does not correspond to any real energy difference. We edited that paragraph in question it to make clearer, see page 8. We stress again, that $\varepsilon = 0.063$ eV and activation energy of 0.15 eV for ion migration measured in [Energy Environ. Sci., 2016, 9, 3180--3187] are two completely different values and cannot be compared.

2. The author proposed that ion migration is the main reason for the PL blinking behavior. In addition, authors mention that the ion migration is photoactivated. It is necessary to explain the detail process, that is, how the photons absorbed by perovskite materials drive the ionic movement, and which is the main driven force. It is noted that, although the illuminated visible light equips with the similar or even higher energy (several eV) than the activation energy of ion migration, these photons cannot directly move these ions.

Reply 3.2.

We thank the reviewer for this concern. It is true that photons cannot directly move the ions. Today several mechanisms were proposed in literature to account for photoactivated ionic transport, e.g. a change of the charge state under illumination and with this a reduced activation barrier for ionic motion (*Energy Environ. Sci.* 2016, 3180), light-induced structural transformations (Zhao et al., *Light: Science & Appl.*, 2016, 6, e16243) or phonon-induced transport (Chen et al., *ACS Appl. Mater. Interfaces* 2016, 8, 5357). However, in our opinion, the origin is not precisely known. In fact, we do not need any photoactivation to explain blinking. We mention it in the discussion only briefly to show that the activation energy for ion migration can be even lower under light illumination. Thus, we believe that suggesting a mechanism for the light activation is not relevant at this point. By referring to the papers of Zhao et al. (Ref. 43) and Chen et al. (Ref. 56) on p. 11, we hope to provide a useful link for further information about different scenarios that might explain light-activated ionic transport.

3. The author cited "Molecules 21, 1–12 (2016).", it should be: "Molecules, 21, 1081 (2018)"

Reply 3.3. The typo has been corrected.

4. Recently, it is proposed that the oxidation state of halide vacancy may play an important role in nonradiative charge recombination. [J. Am. Chem. Soc. 2018, 140, 15753–15763]. It may be helpful to understand the possible PL quenchers in perovskite.

Reply 3.4. We thank the reviewer for this important reference, which we included (Ref. 59) in our revised discussion on the nature of the fluctuating quenchers (see pages 11-12 in the manuscript and more detailed response to the comment of reviewer #1).

REVIEWERS' COMMENTS:

Reviewer #1 (Remarks to the Author):

Authors have further improved their manuscript and well addressed reviewers' concerns. I believe this manuscript provides novel insight into the dynamic interaction between carriers and ions; therefore it can be an helpful reference for deep physical understanding of perovskites and their applications. I recommend publishing this manuscript in Nature Communications.

Reviewer #3 (Remarks to the Author):

In this revised manuscript, authors have made significant improvement on the interpretation. I think authors have reasonably addressed referees' concerns and questions. This manuscript is ready for publication.